# Regional-scale lateral carbon transport and $CO_2$ evasion in temperate stream catchments

Katrin Magin[1], Celia Somlai-Haase[1], Ralf B. Schäfer[1] and Andreas Lorke[1]

[1]Institute for Environmental Sciences, University of Koblenz-Landau, Fortstr. 7, D-76829 Landau, Germany

*Correspondence to:* Katrin Magin (magi6618@uni-landau.de)

**Abstract.** Inland waters play an important role in regional to global scale carbon cycling by transporting, processing and emitting substantial amounts of carbon, which originate mainly from their catchments. In this study, we analyzed the relationship between terrestrial net primary production (NPP) and the rate at which carbon is exported from the catchments in a temperate stream network. The analysis included more than 200 catchment areas in southwest Germany, ranging in size from 0.8 to 889 $km^2$ for which $CO_2$ evasion from stream surfaces and downstream transport with stream discharge were estimated from water quality monitoring data, while NPP in the catchments was obtained from a global data set based on remote sensing. We found that on average 13.9 g C $m^{-2}$ $yr^{-1}$ (corresponding to 2.7% of terrestrial NPP) are exported from the catchments by streams and rivers, in which both $CO_2$ evasion and downstream transport contributed about equally to this flux. The average carbon fluxes in the catchments of the study area resembled global and large-scale zonal mean values in many respects, including NPP, stream evasion as well as the carbon export per catchment area in the fluvial network. A review of existing studies on aquatic-terrestrial coupling in the carbon cycle suggests that the carbon export per catchment area varies in a relatively narrow range, despite a broad range of different spatial scales and hydrological characteristics of the study regions.

**Keywords**

Regional carbon cycle, terrestrial-aquatic coupling, net primary production, $CO_2$ degassing from streams, land use

## 1 Introduction

Inland waters represent an important component of the global carbon cycle by transporting, storing and processing significant amounts of organic and inorganic carbon (C) and by emitting substantial amounts of carbon dioxide ($CO_2$) to the atmosphere (Cole et al., 2007;Aufdenkampe et al., 2011). Globally about 0.32 to 0.8 Pg C is emitted per year as $CO_2$ from lakes and reservoirs (Raymond et al., 2013;Barros et al., 2011). For streams and rivers the global estimates range from 0.35 to 1.8 Pg C $yr^{-1}$ (Raymond et al., 2013;Cole et al., 2007), where the lower estimates can be considered as conservative because they omit $CO_2$ emissions from small headwater streams. In 2015 global $CO_2$ evasion from rivers and streams was estimated at 0.65 Pg C $yr^{-1}$ (Lauerwald et al., 2015). Comparable amounts of carbon are discharged into the oceans by the world's rivers (0.9 Pg C $yr^{-1}$) and stored in aquatic sediments (0.6 Pg C $yr^{-1}$) (Tranvik et al., 2009). In total, evasion, discharge and storage of C in inland waters have been estimated to account for about 4 % of global terrestrial net primary production (NPP) (Raymond et al., 2013) or 50-70 % of the total terrestrial net ecosystem production (NEP) (Cole et al., 2007). A recent continental-scale analysis, which combined terrestrial productivity estimates from a suite of biogeochemical models with estimates of the total aquatic C yield for the conterminous United States (Butman et al., 2015), resulted in mean C export rates from terrestrial into freshwater systems of 4 % of NPP and 27 % of NEP. These estimates varied by a factor of four across 18 hydrological units with surface areas between $10^5$ and $10^6$ $km^2$.

The substantial lateral and vertical transport of terrestrial-derived C in inland waters is currently not accounted for in most bottom-up estimates of the terrestrial uptake rate of atmospheric $CO_2$ (Battin et al., 2009) and results in high uncertainties in regional-scale C budgets and predictions of their response to climate change, land use and water management. Only few studies have quantified C fluxes and pools including inland waters at the regional-scale ($O(10^3$-$10^4$ $km^2$)) (Christensen et al., 2007;Buffam et al., 2011;Jonsson et al., 2007;Maberly et al., 2013) or for small ($O(1$-$10$ $km^2$)) catchments (Leach et al., 2016;Shibata et al., 2005;Billett et al., 2004). The majority of existing regional-scale studies on terrestrial-aquatic C fluxes are from the boreal zone and are characterized by a relatively large fractional surface area covered by inland waters, a high abundance of lakes and high fluvial loads of dissolved organic carbon (DOC). Landscapes in the temperate zone can differ in all these aspects, potentially resulting in differences in the relative importance of aquatic C-fluxes and flux paths (storage, evasion and discharge) in regional-scale C budgets. In this study, we provide a representative investigation of a temperate watershed to improve the understanding of the role of temperate inland water bodies in the regional and global carbon cycles. We analyzed the relationship between terrestrial NPP and $CO_2$ evasion and C discharge for more than 200 catchments in southwest Germany. The stream-dominated catchments range in size from 0.8 to 889 $km^2$ and are characterized by a relatively small fraction of surface water coverage (< 0.5 % of the land surface area). In contrast to studies from the boreal zone, the fluvial C load is dominated by dissolved inorganic carbon (DIC). Estimates of aquatic C export from the catchments were obtained from water quality and hydrological monitoring data and were related to terrestrial NPP derived from MODIS satellite data. The scale dependence of aquatic carbon fluxes in relation to NPP is analyzed by grouping the data according to Strahler stream order (Strahler, 1957). By comparing our results to a variety of

published studies, we finally discuss the magnitude as well as the relative importance of different fluvial flux paths
in regional-scale C budgets in different landscapes and climatic zones.
**2 Materials and Methods**
**2.1 Study area and hydrological characteristics**
The study area encompasses large parts of the federal state of Rhineland-Palatinate (RLP) in southwest Germany
(Fig. 1). The average altitude is 323 m (48 m - 803 m) and the mean annual temperature and precipitation varied
between 5.8 and 12.2 °C and 244 and 1576 mm during the time period between 1991 and 2011 at the 37
meteorological stations operated by the state RLP (http://www.wetter.rlp.de/). The dominating land cover in the
study area is woodland (41 %, mainly mixed and broad-leaved forest), tilled land (37 %, mainly arable land and
vineyards) and grassland (13 %, mainly pastures) (Corine land cover (EEA, 2006)). The fraction of peatland in the
study area is small (0.95 km$^2$; 0.009% of the study area). 16 % of the study area contain carbonate bedrock.
Most of the rivers in RLP are part of the catchment area of the Rhine River. Other large rivers in the state are Mosel,
Lahn, Saar and Nahe. The upland regions of RLP are sources to many small, steep and highly turbulent streams with
gravel beds (MULEWF, 2015). Lakes in RLP are small with a total area of approximately 40 km$^2$ (Statistisches
Landesamt Rheinland-Pfalz, 2014) and were omitted from the analysis. The river network has a total length of
15800 km and consists of stream orders (Strahler, 1957) between 1 and 7. A catchment map of RLP, consisting of
subcatchments of 7729 river segments was provided by the state ministry (MULEWF, 2013), where a river segment
refers to the section between a source and the first junction with another river or between two junctions with other
rivers. All subsequent analyses were conducted separately for each stream order and streams of Strahler order >4
were omitted from the analysis because of the limited sample size with only few catchments available. Moreover,
we omitted streams for which parts of the catchment area were outside of the study area. Overall, 3377, 1619, 861
and 453 stream segments were retained for the analysis for Strahler order 1 to 4, respectively. Annual mean
discharge and length of the river segments were obtained from digital maps provided by the state ministry
(MULEWF, 2013).
**2.2 Aquatic carbon concentrations**
DIC concentrations and partial pressure of dissolved $CO_2$ ($p$CO$_2$) in stream water were estimated from governmental
water quality monitoring data which were acquired according to DIN EN ISO norms (DIN EN ISO 10523:2012-
04;DIN EN ISO 9963-1:1996-02;DIN EN ISO 9963-2:1996-02).. The data include measurements of alkalinity, pH
and temperature which were conducted between 1977 and 2011 (MULEWF, 2013). Sampling intervals differed
between the sites and water sampling was conducted irregularly with respect to year and season. To exclude a
potential bias resulting from the seasonality of DIC concentrations on the analysis, we only considered river
segments for which at least one measurement was available for each season (spring, summer, autumn, winter). From
these measurements, $p$CO$_2$ and DIC concentrations were estimated using chemical equilibrium calculations with the
software PHREEQC (Version 2) (Parkhurst and Appelo, 1999). For 201 river segments with seasonally resolved
measurements, we first computed seasonal mean $pCO_2$ and DIC concentrations, which subsequently were aggregated
to annual mean values averaged over the entire sampling period:
$$\overline{pCO_{2_{annual}}} = (\overline{pCO_{2_{spring}}} + \overline{pCO_{2_{summer}}} + \overline{pCO_{2_{autumn}}} + \overline{pCO_{2_{winter}}})/4 \qquad (1)$$
Measurements of dissolved and total organic C (DOC, TOC) were available only for 64 of these sampling sites.

## 2.3 Estimation of lateral DIC export and catchment-scale $CO_2$ evasion

The lateral export of DIC and the total $CO_2$ evasion from the upstream located stream network was calculated for
each of the 201 sampling sites with seasonally averaged concentration estimates. Lateral DIC export from the
corresponding catchments was calculated as the product of the mean DIC concentration and discharge. $CO_2$ evasion
from the stream network upstream of each sampling site was estimated by interpolating $pCO_2$ for all river segments
without direct measurements by averaging the mean concentrations by stream order and assigning them to all stream
segments of the river network (Butman and Raymond, 2011). Stream width ($w$, in m), depth ($d$, in m) and flow
velocity ($v$, in m s$^{-1}$) were estimated from the discharge ($Q$, in m$^3$ s$^{-1}$) using the following empirical equations
(Leopold and Maddock Jr, 1953):
$$w = a * Q^b \qquad d = c * Q^d \qquad v = e * Q^f, \qquad (2)$$
For the hydraulic geometry exponents and coefficients, the values from Raymond et al. (2012) were used ($b$=0.42,
$d$=0.29, $f$=0.29, $a$=12.88, $c$=0.4 and $e$=0.19).
The water surface area ($A$, in m$^2$) was calculated as the product of length and width of the river segments. The
average slope for each segment was estimated from a Digital Elevation Map (resolution 10 m) provided by the
federal state of Rhineland-Palatinate (LVermGeoRP, 2012). Zhang and Montgomery (1994) investigated the effect
of digital elevation model (DEM) resolution on slope calculation and performance in hydrological models for spatial
resolutions between 2 and 90 m. They found that while a 10-m grid is a significant improvement over 30 m or
coarser grid sizes, finer grid sizes provide relatively little additional resolution. Thus a 10-m grid size represents a
reasonable tradeoff between increasing spatial resolution and data handling requirements for modeling surface
processes in many landscapes. The gas transfer velocity of $CO_2$ at 20°C ($k_{600}$, in m d$^{-1}$) was calculated from slope ($S$)
and flow velocity ($v$, in m s$^{-1}$) (Raymond et al., 2012).
$$k_{600} = S * v * 2841.6 + 2.03 \qquad (3)$$
This gas transfer velocity was adjusted to the in situ temperature ($k_T$, in m d$^{-1}$) using the following equation:
$$k_T = k_{600} * \left(\frac{Sc_T}{600}\right)^{-0.5}, \qquad (4)$$
where $Sc_T$ is the Schmidt number (ratio of the kinematic viscosity of water and the diffusion coefficient of dissolved
$CO_2$) at the in situ temperature (Raymond et al., 2012). Finally the $CO_2$ flux ($F_D$, in g C m$^{-2}$ yr$^{-1}$) for each stream
segment was calculated as:
$$F_D = k_T \cdot K_H (pCO_2 - pCO_{2,a}) \cdot M_C \qquad (5)$$
The partial pressure of $CO_2$ in the atmosphere ($pCO_{2,a}$) was considered as constant (390 ppm) and the Henry
coefficient of $CO_2$ at in-situ temperature ($K_H$ in mol $l^{-1}$ $atm^{-1}$ ) was estimated using the relationship provided in
(Stumm and Morgan, 1996). $M_C$ is the molar mass of C (12 g $mol^{-1}$). Finally, the total $CO_2$ evasion was estimated by
summing up the product of $F_D$ with the corresponding water surface area for all stream segments located upstream
of each individual sampling point.

**2.4 Estimation of the catchment NPP**

Average NPP in the catchment areas of the study sites were obtained from a global data set derived from moderate
resolution imaging spectroradiometer (MODIS) observations of the earth observing system (EOS) satellites, which
is available for the time period 2000 to 2013 with a spatial resolution of 30 arc seconds (~ 1 $km^2$) (Zhao et al., 2005).
In this data set, NPP was estimated based on remote sensing observations of spectral reflectance, land cover and
surface meteorology as described in detail by Running et al. (2004). We used mean NPP data (2000-2013) averaged
over the catchment areas of the individual sampling sites.

**2.5 Statistical analysis**

Linear regressions (F-test) were used to analyze the data. Group differences or correlations with $p<0.05$ were
considered statistically significant. For the regression of total aquatic C export rate and annual catchment NPP, data
were log-transformed to correct for normal distribution. All statistical analyses were performed with R (R
Development Core Team, 2011).

**3 Results**

**3.1 Catchment characteristics and aquatic C load**

The size of the analyzed catchment areas varied over three orders of magnitude (0.8 to 889 $km^2$) and the mean size
increased from 9 $km^2$ for $1^{st}$ order streams to 243 $km^2$ for streams of the order 4 (Table 1). Mean discharge and
catchment area were linearly correlated ($r^2=0.74$, $p<0.001$). The runoff depth, i.e. the stream discharge divided by
the catchment area, was relatively constant across stream orders with a mean value of 0.28 m $y^{-1}$, corresponding to
35 % of the annual mean precipitation rate in the study area. The mean discharge increased more than 30-fold from
0.06 to 2.2 $m^3$ $s^{-1}$ for $1^{st}$ to $4^{th}$ order streams, respectively. Similarly, the estimated water surface area increased with
increasing stream order from 0.24 to 0.42 % of the corresponding catchment size (Table 1).
Individual estimates of the $CO_2$ partial pressure at the sampling sites varied between 145 and 7759 ppm. Only 1 %
of the $pCO_2$ values were below the mean atmospheric value (390 ppm), indicating that the majority of the stream
network was a source of atmospheric $CO_2$ at all seasons. The $pCO_2$ was higher in summer (mean±sd: 2780±2098
ppm) and autumn (mean±sd: 2848±2019 ppm) than in winter (mean±sd: 2287±1716 ppm) and spring (mean±sd:
2172±2343 ppm). The total mean value of $pCO_2$ was 2083 ppm and $pCO_2$ and DIC did not differ significantly
among the different stream orders ($pCO_2$: $p=0.35$; DIC: $p=0.56$). On average, DIC in the stream water was
composed of 91.2 % bicarbonate, 0.4 % carbonate and 8.4 % $CO_2$.

The few available samples of DOC and TOC indicate that the organic C concentration was about one order of
magnitude smaller than the inorganic C concentration (Table 1). There were no pronounced regional or temporal
differences of organic carbon. Only a small fraction of TOC was in particulate form (on average 8.6 %) and TOC
was linearly related to DIC, indicating that the organic load made up only 4 % of the total carbon load at the
sampling sites (Fig. 2). The data are provided as supplementary material.
**3.2 Catchment NPP and C budget**
NPP increased linearly with catchment size ($r^2=0.98$, $p<0.001$), but the specific NPP, i.e. the total NPP within a
catchment divided by catchment area, did not differ significantly ($p=0.24$) among catchments of different stream
orders. The smallest mean value and the largest variability of specific NPP (mean±sd: $466\pm127$ g C m$^{-2}$ yr$^{-1}$, range:
106 to 661 g C m$^{-2}$ yr$^{-1}$) was observed among the small catchments of 1$^{st}$ order streams, while the variability was
consistently smaller for higher stream orders (Table 2). The total average of terrestrial NPP in the study area was
$515\pm79$ g C m$^{-2}$ yr$^{-1}$ (mean±sd).
In a simplified catchment-scale C balance, we consider the sum of the DIC discharge (DIC concentration multiplied
by discharge) measured at each sampling site and the total $CO_2$ evasion from the upstream located stream network
as the total amount of C that is exported from the catchment area through the aquatic conduit. The total evasion was
estimated by interpolation with stream-order specific $pCO_2$ values assigned to the complete stream network. Given
the small number of available measurements, we neglect the fraction of organic C which is exported with stream
discharge. As demonstrated above, TOC load is small in comparison to the DIC load (Fig. 2), resulting in a
comparably small (< 4 %) error.
The resulting $CO_2$ evasion rates decreased slightly, but not significantly ($p=0.26$) for increasing stream orders with a
total mean evasion rate of 2032 g C m$^{-2}$ yr$^{-1}$ (expressed as per unit water surface area) (Table 2). The total aquatic
evasion rate within catchments normalized by the size of the catchment increased significantly with stream order
with a mean value of 6.6 g C m$^{-2}$ yr$^{-1}$. (Table 2).
The total aquatic C export rate, i.e. the sum of evasion and DIC discharge, was strongly correlated with annual mean
NPP averaged over the corresponding catchment area. Linear regression of the log-transformed data results in a
power-law exponent of 1.06, indicating a nearly linear relationship (Fig. 3). As small streams of low stream order
can be directly influenced by local peculiarities, the relationship is more variable for streams of Strahler order 1 and
2, while larger streams represent more average conditions over larger spatial scales with less variability. Most of the
correlation between the total aquatic C export rate and the annual mean NPP, however, can be attributed to their
common linear scale-dependence.

After normalization with catchment area, the total aquatic C export rate increased slightly with stream order (Fig.
4a). Also the ratio of the carbon exported through the aquatic network (i.e. the sum of evasion and discharge) to the
terrestrial NPP, increased slightly, though not significantly ($p$=0.32), from 2.18 % for first-order stream to 2.72 %
for stream order 4 (Fig. 4b). This increase was related to increasing rates of $CO_2$ evasion in streams of higher order
and the contribution of evasion to the total C export rate increased from 39 to 53 % (Fig. 4c). The increasing evasion
is mainly caused by the increasing fractional water surface area for increasing stream orders (Table 1), because the
$CO_2$ fluxes per water surface showed a rather opposing trend with decreasing fluxes for increasing stream orders
(Table 2). On average 1.31 % of the catchment NPP are emitted as $CO_2$ from the stream network and 1.49 % are
discharged downstream (Table 2).

No regional (large-scale) pattern or gradients were observed in the spatial variation of catchment-scale NPP and
aquatic C export (Fig. 5).
**4 Discussion**
**4.1 Uncertainty analysis**
Our estimates are subject to a number of uncertainties associated with sampling and interpolation and systematic
errors including the neglect of carbon burial in sediments, carbon export and evasion as methane and unresolved
spatial and temporal variability.
According to Abril et al. (2015), high uncertainties of $pCO_2$ estimates from pH and alkalinity measurements occur at
pH values <7. In our study, only 7 % of the pH values were <7. For pH>7 the median and mean relative errors are
1% and 15%, respectively (Abril et al., 2015). Raymond et al. (2013) estimated uncertainties from comparisons of
estimates obtained using approaches comparable to the present study with direct measurements of $CO_2$
concentration on streams. For a density of sampling locations of 0.02 sites per $km^2$ (corresponding to this study)
they derived an uncertainty of 30 %. Similarly, Butman and Raymond (2011) estimated uncertainties of overall flux
estimates of 33 %, based on Monte Carlo simulation of similar data for hydrographic units in the United States.
However, we expect that these unbiased, i.e. randomly distributed, uncertainties did not affect the general results of
our model.
While the riverine carbon concentrations were obtained from measurements that covered a time period from 1977 to
2011, the NPP data were available for the time period from 2000 to 2013. In boreal and subtropical rivers a decadal
increasing DIC export due to the climate change and anthropogenic activities has been observed (Walvoord and
Striegl, 2007;Raymond et al., 2008), therefore the different time periods covered by the two data sets might pose a
problem. Comparisons of DIC measurements in the study area between 1977-1999 and 2000-2011 however did not
show significant changes. Furthermore, the sampling frequency for DIC increased so that the majority of DIC
measurements originated from the same time period as the NPP data (Supplementary Material).
The hydraulic geometry exponents and coefficients used in this study were derived from various data sets obtained
in North America, not for central Europe. Unfortunately, we are not aware of a comparably extensive data set of
hydraulic geometry data derived for European rivers. The coefficients have been applied in global studies before,
e.g. Raymond et al. (2013). A comparison of hydraulic geometry coefficients derived from various data sets,
including data from England, Australia and New Zealand, is presented in Butman and Raymond (2011), who
estimated that the error associated with uncertainties of hydraulic geometry coefficients is rather small, compared to
uncertainties derived for C-fluxes.
Carbon burial in sediments was neglected in this study but can make a significant contribution to catchment-scale C
balances. Estimates vary between 22 % at a global scale (Aufdenkampe et al., 2011), 14 % for the Conterminous
U.S. (Butman et al., 2015) and 39% for the Yellow River network (Ran et al., 2015). However, C storage in aquatic
systems occurs mainly in lakes and reservoirs, which are virtually absent in the study area. Therefore we consider
the bias caused by neglecting storage to be small in comparison to remaining uncertainties (30%).
Similarly, the transport of carbon as methane was neglected because measurements of methane concentration or
fluxes were not available for the study area. According to a recent meta-analysis, the dissolved methane
concentration in headwater streams varies mainly between 0.1 and 1 $\mu$mol $L^{-1}$, with streams in temperate forests
being at the lower end (Stanley et al., 2016). As the methane makes up only a small fraction of total carbon in
comparison to the mean DIC concentration in the present study (500 $\mu$mol L-1), it can be assumed that methane
makes a rather small contribution to the catchment scale carbon balance.
Since no time-resolved discharge data were available for the sampling sites we cannot account for extreme events.
Moreover, no information were available if the governmental monitoring included sampling during floods. Given
the stochastic nature and short duration, we expect that such samples are at least underrepresented. Since it has been
observed that high-discharge events can make a disproportionally high contribution to annual mean carbon export
from catchments, we consider our estimates as a lower bound.
**4. 2. An average study region**
The average carbon fluxes in the catchments of the study area resemble global and large-scale zonal mean estimates
in many aspects. The mean atmospheric flux of $CO_2$ from the stream network of $2031\pm1527$ g C $m^{-2}$ $yr^{-1}$ is in close
agreement with bulk estimates for streams and rivers in the temperate zone of 2630 (Aufdenkampe et al., 2011) and
2370 g C $m^{-2}$ $yr^{-1}$ (Butman and Raymond, 2011). The fractional surface coverage of streams and rivers (0.42 % for
stream order 4) corresponds to the global average of 0.47 % (Raymond et al., 2013) and also mean terrestrial NPP in
the catchments (515 g C $m^{-2}$ $yr^{-1}$) was in close correspondence to recent global mean estimates (495 g C $m^{-2}$ $yr^{-1}$
(Zhao et al., 2005)).
By combining $CO_2$ evasion and downstream C-export by stream discharge, we estimated that 13.9 g C $m^{-2}$ $yr^{-1}$,
corresponding to 2.7 % of terrestrial NPP, are exported from the catchments by streams and rivers, in which both
evasion and discharge contributed equally to this flux. Also these findings are in close agreement with global and
continental scale estimates, of 16 and 13.5 g C $m^{-2}$ $yr^{-1}$, respectively (Table 3).

### 4.3. Aquatic C export across spatial scales

Though not exhaustive, Table 3 provides data from a large share of existing studies relating the aquatic C export to terrestrial production in the corresponding catchments which cover a broad range of spatial scales and different landscapes. Except for the tropical forest of the Amazon basin, the aquatic carbon export normalized to catchment area estimated for temperate streams in our study, is surprisingly similar to those estimated at comparable and at larger spatial scale. In the Amazon, the fraction of terrestrial production that is exported by the fluvial network is more than twofold higher (nearly 7 % of NPP (Richey et al., 2002)). However, that a large fraction of the regional NPP in the Amazon is supported by aquatic primary production by macrophytes and carbon export is predominantly controlled by wetland connectivity, with wetlands covering up to 14 % of the land surface area (Abril et al., 2013). An additional peculiarity of the Amazon is, that in contrast to the remaining systems, the vast majority (87 %) of the total C export is governed by $CO_2$ evasion (Table 3), whereas lateral export constitutes a much smaller component. An exceptionally low fraction of NPP that is exported from aquatic systems at larger scale was estimated for the English Lake District (1.6 % (Maberly et al., 2013)), though only $CO_2$ evasion from lake surfaces was considered, i.e. downstream discharge by rivers was ignored. Their estimate agrees reasonably well with the fraction of catchment NPP that was emitted to the atmosphere from the stream network in the present study (1.3 %). If a similar share of catchment NPP was exported with river discharge also in the Lake District, the average mass of C exported from the aquatic systems per unit catchment area would be in close agreement with our and other larger-scale estimates (Table 3).

In more detailed studies at smaller scales and for individual catchments, aquatic C export was exclusively related to net ecosystem exchange (NEE) measured by eddy covariance. Here the estimated fractions of aquatic export range between 2 % of NEE in a temperate forest catchment (only discharge, evasion not considered, (Shibata et al., 2005)) and 160 % of NEE in a boreal peatland catchment (Billett et al., 2004). Analysis of inter-annual variations of stream export from a small peatland catchment in Sweden (Leach et al., 2016) resulted in estimates of C export by the fluvial network between 5.9 and 18.1 g C m$^{-2}$ yr$^{-1}$ over 12 years. The total mean value of 12.2 g C m$^{-2}$ yr$^{-1}$, however, is in close agreement with the present and other larger-scale estimates (Table 3). In contrast to the present study, C export from the peatland catchments were dominated by stream discharge of dissolved organic carbon.

### 4.4 Controlling factors for aquatic C export

We found a significant linear relationship between total catchment NPP and the C export from the catchment in the stream network across four Strahler orders. The relationship was mainly caused by a strong correlation between catchment size and water surface area. As expected for temperate zones, large streams and rivers with large surface area have larger catchments. A study analyzing aquatic carbon fluxes for 18 hydrological units in the conterminous U.S. (Butman et al., 2015) observed a significant correlation between catchment-specific aquatic C yield and specific catchment NEP, which in turn was linearly correlated to NPP. We did not observe such correlation at smaller scale, which could be related to the rather narrow range of variability in NPP among the considered

catchments. Nevertheless, the linear correlation observed by (Butman et al., 2015) indicates that a constant fraction
of terrestrial NPP is exported by aquatic systems if averaged over larger spatial scales.
The relatively narrow range of variability of C export per catchment area (between 9 and 18 g C m$^{-2}$ yr$^{-1}$, with the
two exceptions discussed above) in different landscapes (Table 3) is rather surprising. Although this range of
variation is most likely within the uncertainty of the various estimates, the variability across different landscapes is
certainly small in comparison to the order of magnitude differences in potential controlling factors like catchment
NPP, fractional water coverage as well as size and climatic zone of the study area. In lake-rich regions, evasion from
inland waters was observed to be dominated by lakes (Buffam et al., 2011;Jonsson et al., 2007), which cover up to
13 % of the surface area of these regions. In the present as well as in other studies on catchments where lakes are
virtually absent (Wallin et al., 2013) and the fractional water coverage was smaller than 0.5 % of the terrestrial
surface area, an almost identical catchment-specific C export and evasion rate has been observed (Table 3). $CO_2$
emissions from water surfaces depend on the partial pressure of $CO_2$ in water and are therefore related to DIC,
which was the dominant form of dissolved C in the present study. Studies in the boreal zone, where dissolved C in
the aquatic systems is mainly in the form of DOC, however, found comparable catchment-specific C export and
evasion rates ((Leach et al., 2016;Jonsson et al., 2007;Wallin et al., 2013), cf. Table 3). The difference in the
speciation of the exported C indicates that a larger fraction of the terrestrial NPP is respired by heterotrophic
respiration in soils and exported to the stream network as DIC in the present study, in contrast to export as DOC and
predominantly aquatic respiration. Observations and modeling of terrestrial-aquatic C fluxes across the U.S.
suggested a transition of the source of aquatic $CO_2$ from direct terrestrial input to aquatic $CO_2$ production by
degradation of terrestrial organic carbon with increasing stream size (Hotchkiss et al., 2015). Such transition was not
observed in the present study, where organic carbon made a small contribution to the fluvial carbon load across all
investigated stream orders. In addition to soil respiration, mineral weathering also contributes to DIC in stream
water. The relative importance of soil respiration and weathering varies depending on geology and the presence of
wetlands in the area (Hotchkiss et al., 2015;Lauerwald et al., 2013;Jones et al., 2003). In the present study, 16 % of
the catchment areas contained carbonate bedrock. The DIC concentration in the water increased with the proportion
of carbonate containing bedrock in the catchment ($R^2$=0.33, $p$<0.001).
Despite the small number of observations in the meta-analysis, the narrow range of variability of C export per
catchment area may indicate that neither water surface area nor the location of mineralization of terrestrial derived C
(soil respiration and export of DIC versus export of DOC and mineralization in the aquatic environment), are
important drivers for the total C export from catchments by inland waters at larger spatial scales. This rather
unexpected finding deserves further attention, as it suggests that other, currently poorly explored, processes control
the aquatic-terrestrial coupling and the role of inland waters in regional C cycling. Given the significant contribution
of inland waters to regional and global scale greenhouse gas emissions, the mechanistic understanding of these
processes is urgently required to assess their vulnerability to ongoing climatic and land use changes, as well to the
extensive anthropogenic influences on freshwater ecosystems. Recent developments of process-based models, which
are capable of resolving the boundless biogeochemical cycle in the terrestrial–aquatic continuum from catchment to
continental scales (Nakayama, 2016), are certainly an important tool for these future studies.

**5 Conclusion:**

Our analysis of the carbon budget in a temperate stream network on regional scale revealed a relationship of aquatic
carbon export and terrestrial NPP. On average 13.9 g C m$^{-2}$ yr$^{-1}$, corresponding to 2.7 % of the terrestrial NPP, were
exported from the catchments by rivers and stream with $CO_2$ evasion and downstream transport contributing equally
to the export. A comparison of our regional scale study with other studies from different scales and landscapes
showed a relatively narrow range of variability of carbon export per catchment area. Future research is needed to
understand the processes that control the aquatic-terrestrial coupling and the role of inland waters in regional carbon
cycling.

**Acknowledgments**
This study was financially supported by the German Research Foundation (grant no. LO 1150/9-1). We thank
Miriam Tenhaken for contributing to a preliminary analysis. All raw data for this paper is properly cited and referred
to in the reference list. The processed data, which were used to generate the figures and tables, are available upon
request through the corresponding author.

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

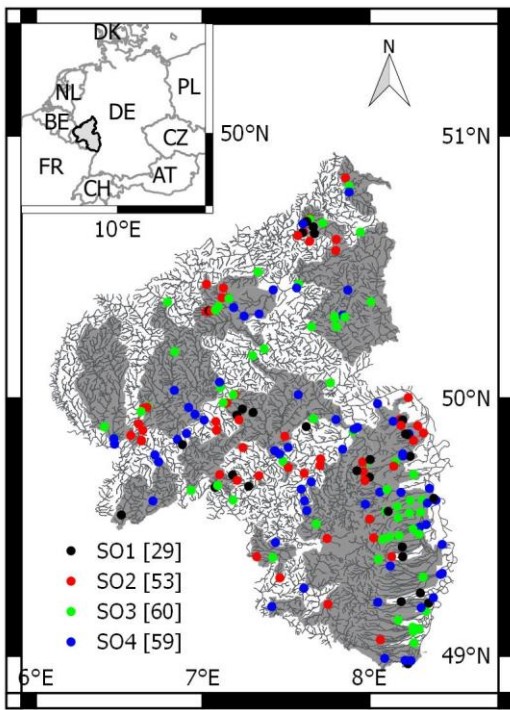


**Fig. 1: Map of the stream network (black lines) within the state borders of Rhineland Palatinate in southwest Germany.**
**The inset map in the upper left corner indicates the location of the study region in central Europe. Filled circles mark the**
**position of sampling sites with color indicating stream order (SO1 – SO4; the numbers in brackets in the legend are the**
**respective number of sampling sites). The catchment areas of the sampling sites are marked in grey color.**

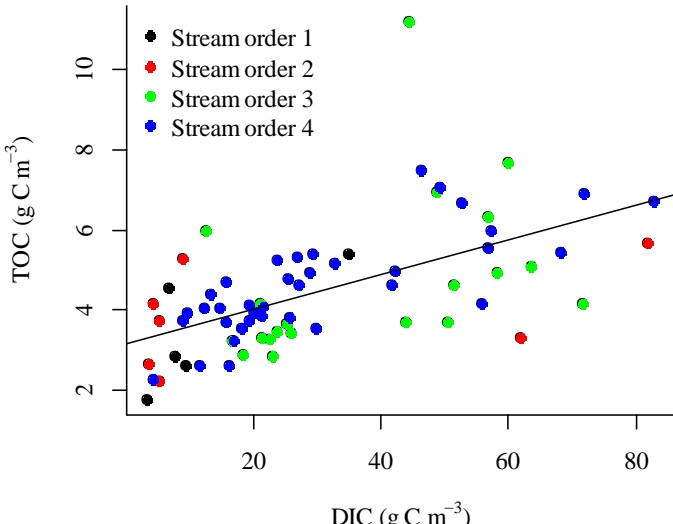


**Fig. 2: TOC concentration versus DIC concentration. Different colors indicate sampling sites from different stream orders. The solid line shows the fitted linear regression model with TOC=0.04·DIC ($r^2$=0.33, $p$<0.001).**


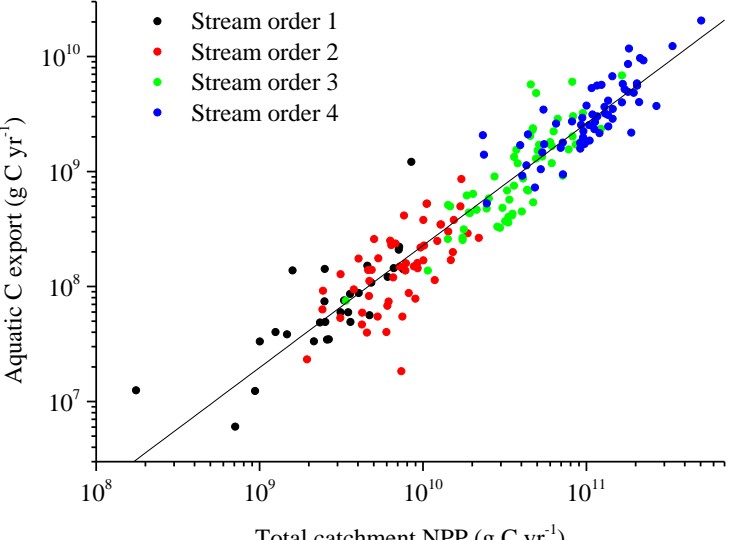


**Fig. 3: Annual rate of C export through the stream network versus terrestrial NPP in the catchment area. Different colors indicate sampling sites from different stream orders. The solid line shows the fitted linear regression model for the log-transformed data with C_export=0.005·NPP$^{1.06}$ ($r^2$=0.89, $p$<0.001).**


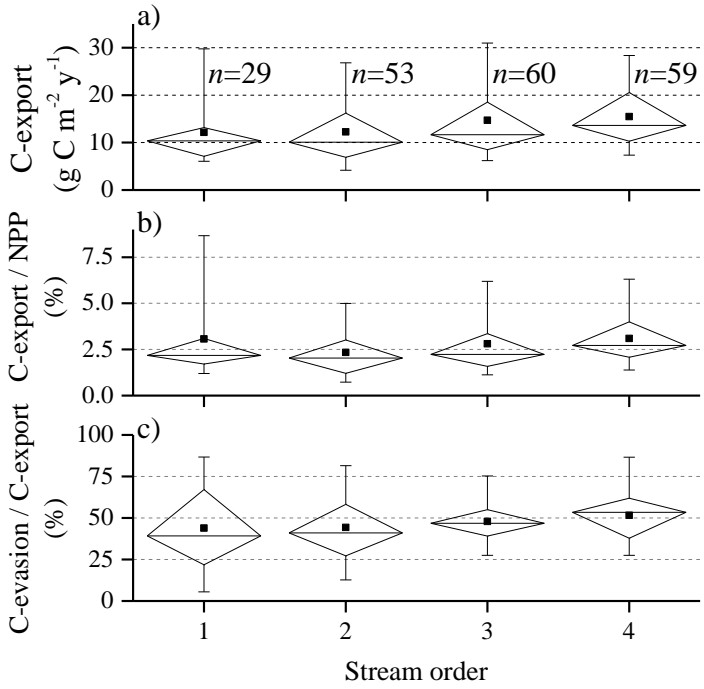


Fig. 4: a) Boxplots of C export (sum of evasion and discharge) normalized by catchment area. b) Boxplots of the ratio of the total exported C and terrestrial NPP for different stream orders. c) Boxplots of the fraction of the total exported C which is emitted to the atmosphere from the stream network for each stream order. The boxes demarcate the 25th and 75th percentiles, the whiskers demarcate the 95% confidence intervals. Median and mean values are marked as horizontal lines and square symbols, respectively. The sample numbers ($n$) provided in a) apply to all panels.


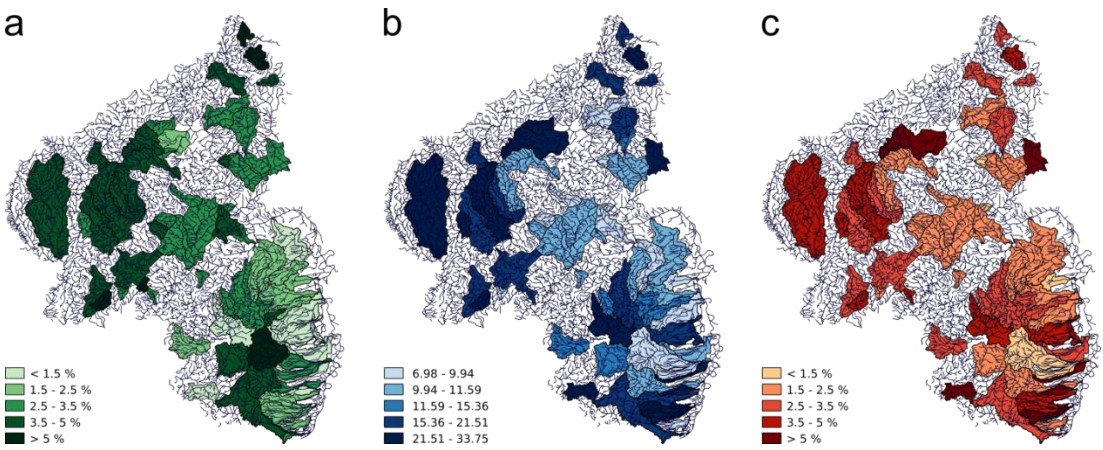


Fig. 5: Map of 3rd and 4th order catchments showing a) Mean NPP (g C m$^{-2}$ yr$^{-1}$), b) aquatic export (g C m$^{-2}$ yr$^{-1}$), c) ratio aquatic export/NPP (%).


**Table 1: Major hydrological characteristics, $pCO_2$, DIC and DOC concentrations averaged over stream orders (SO) and for all sampling sites (total). All values are provided as mean±sd (standard deviation) of the annual mean observations, ranges are given in brackets, $n$ is the number of observations.**

| | SO 1 | SO 2 | SO 3 | SO 4 | Total |
|---|---|---|---|---|---|
| $n$ | 29 | 53 | 60 | 59 | 201 |
| Catchment size (km$^2$) | 9±7 (1 – 35) | 16±9 (4 – 37) | 87±54 (9 – 298) | 243±140 (48 – 889) | 103±126 (1 – 889) |
| Water coverage (%) | 0.24±0.11 (0.05 – 0.43) | 0.26±0.09 (0.1 – 0.45) | 0.36±0.11 (0.09 – 0.6) | 0.42±0.13 (0.18 – 0.7) | 0.33±0.13 (0.05 – 0.7) |
| Discharge (m$^3$ s$^{-1}$) | 0.06±0.05 (0.003 – 0.19) | 0.15±0.10 (0.01 – 0.36) | 0.73±0.63 (0.02 – 3.41) | 2.20±1.95 (0.22 – 12.22) | 0.91±1.41 (0.003 – 12.22) |
| Drainage rate (m y$^{-1}$) | 0.26±0.17 (0.05 – 0.67) | 0.29±0.16 (0.06 – 0.66) | 0.27±0.17 (0.05 – 0.74) | 0.30±0.21 (0.06 – 1.20) | 0.28±0.18 (0.05 – 1.20) |
| pH | 7.58±0.61 (6.20 – 8.97) | 7.70±0.46 (6.30 – 8.60) | 7.81±0.37 (6.60 – 8.30) | 7.75±0.29 (6.91 – 8.30) | 7.73±0.42 (6.20 – 8.97) |
| Alkalinity (mmol L$^{-1}$) | 3.08±2.50 (0.08 – 7.58) | 2.74±2.58 (0.08 – 8.55) | 2.77±1.85 (0.14 – 9.88) | 2.58±1.73 (0.32 – 7.22) | 2.75±2.12 (0.08 – 9.88) |
| $pCO_2$ (ppm) | 2597±1496 (145 – 6706) | 1819±1095 (681 – 5338) | 1992±1327 (573 – 7627) | 2162±1302 (366 – 7759) | 2083±1303 (145 – 7759) |
| DIC (g m$^{-3}$) | 38.8±30.3 (3.4 – 93.1) | 34.2±31.1 (3.5 – 104.5) | 34.6±22.4 (3.1 – 119.6) | 32.4±21.0 (4.1 – 89.3) | 34.5±25.7 (3.1 – 119.6) |
| DOC (g m$^{-3}$) | 3.54±1.86 (2.2 – 6.7) ($n$=5) | 4.11±0.73 (3.1 – 4.8) ($n$=4) | 4.17±1.08 (2.6 – 7.1) ($n$=22) | 4.10±1.24 (2.0 – 7.7) ($n$=33) | 4.08±1.20 (2.0 – 7.7) ($n$=64) |


**Table 2: Aquatic C-fluxes and terrestrial NPP in catchments drained by streams of different stream orders (SO) and for all sampling sites (total). All values are mean ± standard deviation, ranges are given in brackets. The $CO_2$ flux from the water surface (first row) is expressed per square meter water surface area, while the remaining fluxes are expressed per square meter catchment area.**

| | SO 1 | SO 2 | SO 3 | SO 4 | Total |
|---|---|---|---|---|---|
| $CO_2$ flux from water surface (g C m$^{-2}$ yr$^{-1}$) | 2415±2335 (-335 – 12915) | 1975±1364 (418 – 7143) | 1998±1671 (704 – 11016) | 1928±903 (851 – 5093) | 2032±1528 (-335 – 12915) |
| Gas transfer velocity k600 (m d$^{-1}$) | 7.04±4.52 (2.16 – 20.57) | 7.74±3.78 (2.03 – 20.50) | 5.86±2.81 (2.03 – 15.55) | 4.23±0.96 (2.03 – 6.50) | 6.05±3.32 (2.03 – 20.57) |
| $CO_2$ evasion per catchment area (g C m$^{-2}$ yr$^{-1}$) | 5.9±6.3 (-1.0 – 30.0) | 5.2±4.1 (0.7 – 19.2) | 7.0±6.6 (1.6 – 43.8) | 8.0±4.6 (3.0 – 23.0) | 6.6±5.5 (-1.0 – 43.8) |
| DIC discharge per catchment area (g C m$^{-2}$ yr$^{-1}$) | 6.2±4.5 (1.6 – 25.8) | 7.1±6.1 (0.6 – 27.2) | 7.7±5.7 (1.6 – 35.5) | 7.5±4.7 (1.2 – 24.5) | 7.3±5.4 (0.6 – 35.5) |
| Total aquatic C export per catchment area (g C m$^{-2}$ yr$^{-1}$) | 12.1±6.9 (4.7 – 34.5) | 12.3±6.9 (1.5 – 29.6) | 14.7±10.8 (5.3 – 66.8) | 15.5±6.7 (7.0 – 33.8) | 13.9±8.3 (1.5 – 66.8) |
| NPP (g C m$^{-2}$ yr$^{-1}$) | 466±127 (106 – 661) | 536±66 (251 – 644) | 527±57 (364 – 627) | 508±69 (330 – 618) | 515±79 (106 – 661) |

522

**Table 3: Summary of estimates of aquatic C export in relation to terrestrial production in the watershed across different spatial scales (spatial scale decreases from top to bottom). Aquatic C export is the sum of C-discharge and evasion (numbers in parentheses also include the change in C storage in the aquatic systems by sedimentation) normalized by the area of the terrestrial watershed. Aquatic C fate refers to the percentage of the total exported C which is emitted to the atmosphere (evasion) and transported downstream (discharge). The missing percentage is the fraction which is stored in the aquatic systems by sedimentation (if considered). Terrestrial production is expressed as NPP or as net ecosystem exchange (NEE). n.c. indicates that this compartment/flux was not considered in the respective study.**

| Study area (Catchment size in km$^2$) | Fractional water coverage (%) Rivers Lakes | Aquatic C export (g C m$^{-2}$ yr$^{-1}$) | Aquatic C fate (%): Evasion Discharge | Aquatic C export / terrestrial production (%) NPP | NEE | Reference |
|---|---|---|---|---|---|---|
| Global (1.3x10$^8$) | R: 0.2-0.3 L: 2.1-3.4 | 16 (20) | E: 44 D: 34 | 3.7 [1] | 21-64 [2] | (Aufdenkampe et al., 2011) |
| Conterminous U.S. (7.8x10$^6$) | R: 0.52 L: 1.6 | 13.5 (18.8) | E: 58 D: 28 | 3.6 | 27 [3] | (Butman et al., 2015) |
| Central Amazon (1.8x10$^6$) | 4-16 | 138 | E: 87 D: 13 | 6.8 [4] | n.c. | (Richey et al., 2002) |
| Yellow River network (7.5x10$^5$) | R: 0.3-0.4 L: n.c. | 18.5 (30) | E: 35 D: 26 | n.c. | 96 (62) | (Ran et al., 2015) |
| North temperate | R: 0.5 | 11.8 | E: 33 | n.c. | 7 | (Buffam et al., |

| | | | | | | |
|---|---|---|---|---|---|---|
| lake district (6400) | L: 13 | (16) | D: 41 | | | 2011) |
| Northern Sweden (peat) (3025) | R: 0.33 <br> L: 3.5 | 9 | E: 50 (4.5) <br> D: 50 (4.5) | n.c. | 6 | (Jonsson et al., 2007) |
| **Temperate streams (0.7- 1227)** | **R: 0.33** <br> **L: n.c.** | **13.9** | E: 47 <br> D: 53 | **2.7** | n.c. | **This study** |
| English Lake district (1 - 360) | R: n.c. <br> L: 2.2 | 5.4 | E: 100 <br> D: n.c. | 1.6 | n.c. | (Maberly et al., 2013) |
| Forested stream catchments in Sweden (0.46 - 67) | R: 0.1-0.7 <br> L:n.c. (<0.7) | 9.4 | E: 53 <br> D: 47 | n.c. | 8-17 | (Wallin et al., 2013) |
| Forest catchment in Japan (9.4) | R: - <br> L: n.c. | 4 | E: n.c. <br> D: 100 | n.c. | 2 | (Shibata et al., 2005) |
| Peatland catchment (3.35) | R: 0.05 <br> L: n.c. | 30.4 | E: 13 <br> D: 87 | n.c. | 160 | (Billett et al., 2004) |
| Peatland catchment (2.7) | R: n.c. <br> L: 2.2 | 12.2 | E: - <br> D: - | n.c. | 12-50 | (Leach et al., 2016) |

[1] For a value of 56 Pg C yr$^{-1}$ for global NPP (Zhao et al., 2005).
[2] Global mean NEE was estimated as the difference of GPP and ecosystem respiration, which was assumed to be 91-
97 % of GPP (Randerson et al., 2002).
[3] This percentage refers to NEP instead of NEE.
[4] For a global mean value of NPP in tropical forests of 1148 g C m$^{-2}$ yr$^{-1}$ (Sabine et al., 2004).