# Peer review of "Regional-scale lateral carbon transport and CO2 evasion in temperate stream catchments"

_Biogeosciences, 2017_

## Referee Comment (RC1) · Anonymous Referee #1 · 8 Mar 2017

General Comments

Rivers and streams are an important link in the global C cycle and C export via aquatic systems has repeatedly been concluded to make a significant proportion of catchment C budgets at different spatial scales and in different climate zones. This is an interesting paper that will make a good contribution to the understanding of regional-scale carbon export via streams with stream order 1-4 in the temperate zone.

The authors observed a narrow range of variability of C export per catchment area and conclude that other processes than water surface area or location of mineralization of terrestrial derived C control the aquatic-terrestrial coupling and the role of inland waters in regional C cycling. However, the final version of the paper would benefit from more details about lateral C export calculations. It is not clear if extreme runoff events are

covered appropriately. The lack of extreme event data would of course lead to a much narrower range of variability of C export.

Specific Comments

Ln 20. Explain "catchment-specific"

Ln 53. Why is the fluvial C load dominated by DIC? Are there carbonates? Please also state why you neglected methane.

Ln 56. Strahler stream order?

Ln 61-66. What about the geology? Is there C-containing bedrock in the catchments?

Ln 70. "15 800" do not separate numbers

Ln 71. Delete "order"

Ln 82. How are pH values of investigated waters? The pCO2 calculation with alkalinity was found to high uncertainties for low pH values (Abril et al. 2014).

Ln 89. How exactly did you aggregate annual means? Did you calculate a (discharge) weighted average?

Ln 128. Name the program used for statistics

Ln 145-148. Discuss variance of organic C. How about peaty areas?

Ln 152. Specify which value is meant: mean NPP or mean specific NPP?

Ln 169. In Figure 3 some of the data points (mostly stream order 1 and one of stream order 2) scatter more. Please discuss reasons for these outliers.

Ln 186. You talk about average fluxes, but what happens during floods/ extreme events? Do measurement intervals cover extreme events?

Ln 205. "…wetlands covering up to 16 % of the land surface area." Add a reference.

[Figure]

Ln 226. Expected for temperate zones? In dry regions such as deserts this can be different.

Ln 235. Discuss "uncertainty of the various estimates"

Ln 236. Name potential controlling factors

Ln 244-247. How do you know that? In regions with corresponding geology also weathering of C-bearing minerals can be a large source of stream DIC. Respiration in soils is more likely the dominant DIC source in catchments that lack carbonate rocks. Is that true for catchments in Rhineland-Palatinate? Can you give an example for cases with predominance of aquatic respiration? I would expect predominance of aquatic respiration in warmer climates where large DOC concentrations prevail.

Ln 249. How is the range of discharge? The study by Hotchkiss et al. covers values from 0.0001 to 10,000 m3 s-1. Can the lower range in your study be the reason that you do not observe findings in Hotchkiss et al.?

Ln 252. Does "small number of observations" relate to this study?

Ln 255-257. This section summarizes the paper well but it could go further. It might be speculative but can you say what these other, poorly explored processes could be?

Ln 267. I think it is preferable to provide data as supplement material.

Table 1 and Table 2 would be more informative if you could add ranges. Please also add calculated gas transfer velocity values to Table 2.

References:

Abril, G., S. Bouillon, F. Darchambeau, C. R. Teodoru, T. R. Marwick, F. Tamooh, F. O. Omengo, N. Geeraert, L. Deirmendjian, P. Polsenaere, and A. V. Borges (2015), Technical Note: Large overestimation of pCO2 calculated from pH and alkalinity in acidic, organic-rich freshwaters, Biogeosciences, 12(1), 67-78, doi:10.5194/Bg-12-67-2015.

---

## Referee Comment (RC2) · Anonymous Referee #2 · 11 Mar 2017

Accurate estimation of aquatic carbon export is essential to understand the role of natural ecosystems and geochemical processes in global carbon cycles in the context of climate change and increasing anthropogenic activities. In this manuscript, the authors integrate the analysis of downstream export of riverine carbon and CO2 evasion to the atmosphere from more than 200 local catchments of variable sizes in temperate Europe along with the model estimation of ecosystem production. Based on this large dataset, the authors try to establish a carbon budget in a local scale and discuss the ecologic factors controlling the aquatic carbon export. Overall, the integration of the large dataset of riverine carbon concentrations spanning over last several decades is technically sound and strengthens the arguments in the manuscript.

My biggest concern arises from the estimation of the downstream export of riverine carbon. The riverine carbon concentrations adopted in this investigation were obtained

during 1977-2011, which is significantly longer than NPP of 2000-2013. Investigations have already showed a decadal increasing DIC export in boreal and subtropical rivers due to the climate change and anthropogenic activities (Walvoord, M. A., and R. G. Striegl, 2007, Increased groundwater to stream discharge from permafrost thawing in the Yukon River basin: Potential impacts on lateral export of carbon and nitrogen, Geophys. Res. Lett., 34, L12402, doi:10.1029/2007GL030216; Raymond, P.A., Oh, N.-H., Turner, R.E., Broussard, W., 2008. Anthropogenically enhanced fluxes of water and carbon from the Mississippi River. Nature 451, 449-452). Therefore, I would suggest using the environment monitoring dataset during the last 10 years or so, which is consistemt with NPP estimation, to estimate the riverine carbon export.

Secondly, it seems that the data points for the flux estimation is sparse as indicated in the section 2.2 (see Page 3 Line 83-86: ". . . . . .at least one measurement was available for each season. . . . . .."). Therefore, I will be happy to see the error or uncertainty analysis of the flux estimation with the method using the mean concentration and total river discharge (see Page 4 Line 94-95). Moreover, a comparison with other flux estimation methods, such as the one using flow-weighted mean concentration and discharge, the one based on the regression of instantaneous flux and discharge, and other methods (see Warnken, K.W., Santschi, P.H., 2004. Biogeochemical behavior of organic carbon in the Trinity River downstream of a large reservoir lake in Texas, USA. Sci. Total Environ. 329, 131-144), will be helpful to validate the flux estimation.

What do you mean "interpolating $pCO_2$ for all river segments without direct measurement" (Page 4 Line 95-97)? Please clarify in the text.

For DOC, there are 64 observations (Table 1) in 54 sampling sites (Page 3 Line 90-91). On average, there are less than 2 observations in each site. Usually, DOC concentrations in rivers could vary seasonally with river discharge by couples of times. Therefore, the representativeness of the single DOC data in each catchment remains a critical question which may induce the great deviation of DOC flux estimation from the real value. Before resolving this issue, the statements that DOC load only made up

4% total carbon load (Page 5 Line 146-148) and that the error would be comparably small when neglecting the DOC term (Page 6 Line 159-162) seem arbitrary.

The authors extensively discuss the aquatic carbon export/NPP ratio in the manuscript (See Table 3 and text in Section 4.1s). They state in the manuscript:" By combining $CO_2$ evasion and downstream C-export by stream discharge, we estimated that 2.7 % of terrestrial NPP (13.9 g C m2 yr-1) are exported from the catchments by streams and rivers, in which both evasion and discharge contributed equally to this flux (Page 7 Line 193-195)". Then they compare their results with some other studies of catchment ecosystems (see text in Section 4.2). However, what I understand is riverine DIC export flux is closely related to the weathering regimes and intensity in catchments (See Cai, W.-J., Guo, X., Chen, C.-T.A., Dai, M., Zhang, L., Zhai, W., Lohrenz, S.E., Yin, K., Harrison, P.J., Wang, Y., 2008. A comparative overview of weathering intensity and HCO3- flux in the world's major rivers with emphasis on the Changjiang, Huanghe, Zhujiang (Pearl) and Mississippi Rivers. Continental Shelf Research 28, 1538-1549; and Raymond, P.A., Bauer, J.E., Caraco, N.F., Cole, J.J., Longworth, B., Petsch, S.T., 2004. Controls on the variability of organic matter and dissolved inorganic carbon ages in northeast US rivers. Marine Chemistry 92, 353-366) although NPP could contribute part of DIC export flux through the respiration of DOM. Therefore, the aquatic carbon export/NPP ratio would be expected to be larger than the real contribution of NPP.

---

## Referee Comment (RC3) · Anonymous Referee #1 · 20 Mar 2017

Please check equation 1. The exponents and coefficients between width and depth are switched. In "Stream Hydraulics" in Raymond et al. (2012) this can be checked. I think the correct formulas would be w=12.88*Qˆ0.42 and d=0.4*Qˆ0.29. This corresponds to width=c*Qˆd and depth=a*Qˆb. The coefficients and exponents of your equation lead to depth»width, which seems not realistic.
* * *

---

## Referee Comment (RC4) · Anonymous Referee #3 · 21 Mar 2017

General comments:

Katrin Magin and colleagues presented a synthesis of >200 catchments examining the relationships between lateral carbon export and CO2 emissions and terrestrial net primary production (NPP) in southwest Germany. Inland waters have recently been recognized as important components in the global carbon cycle. While widespread studies have been conducted worldwide, most of these studies are based on individual catchments and a synthesis involving multiple catchments remains lacking. This manuscript is well-organized and quite timely, and will provide insights into the understanding of catchment carbon cycle (or budget) at regional scales.

My first major concern after reading this manuscript is the carbon storage term which has not yet been considered when the authors evaluated catchment-scale carbon budget. Caron burial associated with soil erosion and sediment deposition within catchments is a quite important component in carbon budget assessments (e.g., Smith et al., 2001). If the traditional sediment delivery ratio of 10% is assumed (Harden et al., 1999), 90% of the eroded POC from land may have been stored somewhere within the catchment and partly exposed to decomposition (thus evasion to the atmosphere). This missing term may affect the redistribution of carbon (downstream discharge vs. CO2 evasion) as well as the amount of total carbon input from land. Incorporating this term will thus refine the budget result.

My second concern is the estimation of CO2 evasion. What are the resulting k600 values? Are they comparable to those based on field direct measurements (e.g., floating chamber or eddy covariance)? Estimation of the total areal extent of water surface by means of the parameters derived from USA catchments is probably problematic (see my specific comment below). In addition, can the available dataset suggest any seasonal variability in CO2 evasion?

Specific comments:

Line 19: please clarify 'catchment-specific total export rate'. Is it the normalized carbon export by catchment area?

Line 29-30: the latest CO2 evasion from global rivers and streams is 0.65 Pg C/yr by Lauerwald et al., (2015).

Line 50: remove 'differ'.

Line 71: the reference 'Strahler, 1957' should move to line 56.

Line 77: remaining→retained

Line 81-83. What's the data quality and what kinds of standards for water sampling and processing were used? Estimating pCO2 from alkalinity and pH has been criticized for causing biases due to noncarbonate impacts (Abril et al., 2015). An uncertainty analysis should be provided here. I also suggest to provide the range of pH and alkalinity,

possibly into Table 1.

Line 95-97: how was the site-specific pCO2 interpolated to the upstream catchments? And which interpolation technique was used?

Line 102-103: These arbitrary parameters derived from American rivers may not necessarily be representative of German rivers. See Leopold and Maddock (1953).

Line 105: Is a resolution of 10m enough to estimate channel slope changes?

Line 125-126: Because the mean NPP for the period 2000-2013 is used here while the pCO2 data is for the period 1970-2011, it is better to explicitly indicate the distribution frequency of pCO2 data over the study period. For example, if the most of the pCO2 data were for the period 1970-1980, then using the NPP for 2000-2013 would be problematic.

Line 135: Based on the given definition, the 'drainage rate' term should be 'runoff depth' in a formal way.

Line 146: Please quantify 'only a small fraction'.

Line 158: For the total C input, how about the POC term and the carbon storage term? See my major comment.

Line 220-223: Are there peatlands within the studied catchments?

Line 229-230: Is the absence of the carbon yield and NPP correlation due to failure to measure pCO2 during flooding periods? The short-duration carbon export during flooding events usually accounts for disproportionately a large share of the annual total carbon export.

Line 238: please clarify the 'surface area'. The global surface area?

Line 244-247: Could it also be because of chemical weathering and groundwater inputs? Rock weathering in carbonate-dominated catchments can be a significant contributor to DIC. I would suggest the authors to make a brief introduction about the lithology and mineralogy in the study area section (2.1).

Line 262: please summarize the study and make a short conclusion.

Figure 2. It seems the top 2(?) data points far away from the majority are outliers. Please check and make the regression again, if necessary.

Table 1. pH could also be tabulated here. Is there any trend in pH from SO1 to SO4?

References

Abril, G., Bouillon, S., Darchambeau, F., Teodoru, C., Marwick, T., Tamooh, F., Omengo, F., Geeraert, N., Deirmendjian, L., Polsenaere, P., and Borges, A. V.: Technical Note: Large overestimation of pCO2 calculated from pH and alkalinity in acidic, organic-rich freshwaters, Biogeosciences, 12, 67-78, 2015.

Harden, J., Sharpe, J., Parton, W., Ojima, D., Fries, T., Huntington, T., and Dabney, S.: Dynamic replacement and loss of soil carbon on eroding cropland, Global Biogeochem. Cy., 13, 885–901, 1999.

Lauerwald, R., Laruelle, G. G., Hartmann, J., Ciais, P., and Regnier, P. A.: Spatial patterns in CO2 evasion from the global river network, Global Biogeochemical Cycles, 29, 534-554, 2015.

Leopold, L., and Maddock, T.: The hydraulic geometry of stream channels and some physiographic implications, USGS Professional Paper 252, USGS Professional Paper 252, 1953.

Smith, S. V., Renwick, W. H., Buddemeier, R. W., and Crossland, C. J.: Budgets of soil erosion and deposition for sediments and sedimentary organic carbon across the conterminous United States, Global Biogeochemical Cycles, 15, 697-707, 2001.

---

## Author Comment (AC1) · 28 Apr 2017

Anonymous Referee #1 :

- Rivers and streams are an important link in the global C cycle and C export via aquatic systems has repeatedly been concluded to make a significant proportion of catchment C budgets at different spatial scales and in different climate zones. This is an interesting paper that will make a good contribution to the understanding of regional-scale carbon export via streams with stream order 1-4 in the temperate zone. The authors observed a narrow range of variability of C export per catchment area and conclude that other processes than water surface area or location of mineralization of terrestrial derived C control the aquatic-terrestrial coupling and the role of inland waters in regional C cycling. However, the final version of the paper would benefit from more

details about lateral C export calculations. It is not clear if extreme runoff events are covered appropriately. The lack of extreme event data would of course lead to a much narrower range of variability of C export.

We would like to thank the reviewer for her/his positive evaluation and the very helpful comments and suggestions. Below we reply on each specific comment.

- Ln 20. Explain "catchment-specific"

Reply: Catchment-specific carbon export refers to the carbon export per catchment area. We will explain this in a revised version of the manuscript.

- Ln 53. Why is the fluvial C load dominated by DIC? Are there carbonates? Please also state why you neglected methane.

Reply: 16% of the study area contain carbonate bedrock. The weathering of the carbonate rock can be an additional source of DIC in the streams. In our study area, the DIC concentration in the water increased with the proportion of carbonate containing bedrock in the catchment (R2=0.33, p<0.001). We would add the information on bedrock in a revised version of the manuscript. On average, DIC in the stream water was composed of 91.2% bicarbonate, 0.4% carbonate and 8.4% CO2. Alkalinity ranged between 0.02 – 13.5 mmol L-1 for the individual measurements and between 0.08 – 9.88 mmol L-1 for the averaged seasonal mean values. Measurements of methane concentration or fluxes are not available for the present study. According to a recent meta-analysis, the dissolved methane concentration in headwater streams varies mainly between 0.1 and 1 $\mu$mol L-1, with streams in temperate forests being at the lower end (Stanley et al., 2016). The methane makes up only a small fraction of total carbon (in comparison to the mean DIC concentration in the present study (500 $\mu$mol L-1)) and we assume that methane makes a rather small contribution to the catchment scale carbon balance. We would add these information, together with an upper bound of methane evasion (based on the published meta-analysis), to the revised manuscript.

- Ln 56. Strahler stream order?

Reply: Yes, Strahler stream order. We will add this information to the revised manuscript.

- Ln 61-66. What about the geology? Is there C-containing bedrock in the catchments?

Reply: See our comment above - 16% of the study area contain carbonate bedrock. The weathering of the carbonate rock can be one additional source of DIC in the streams. In our study area, the DIC concentration in the water increased with the proportion of carbonate containing bedrock in the catchment ($R2=0.33$, $p<0.001$). We would add the information on the bedrock in a revised version of the manuscript.

- Ln 70. "15 800" do not separate numbers

- Ln 71. Delete "order"

Reply: We will correct this.

- Ln 82. How are pH values of investigated waters? The $pCO2$ calculation with alkalinity was found to high uncertainties for low pH values (Abril et al. 2014).

Reply: The range of pH values of the investigated waters is 6.2 – 8.97 with a mean of 7.73±0.42 (mean±sd). According to (Abril et al., 2015), high uncertainties of $pCO2$ estimates from pH and alkalinity measurements occur at pH values <7, while the median and mean relative errors were 1% and 15%, respectively for pH>7. Only 7% of the pH values in our study were <7. We would add a discussion of the expected uncertainties to a revised manuscript.

- Ln 89. How exactly did you aggregate annual means? Did you calculate a (discharge) weighted average?

Reply: $pCO2\_annual=(pCO2\_spring+pCO2\_summer+pCO2\_autumn+pCO2\_winter)/4$ (seasonal values are averaged over all available samples). Discharge was not measured during the water sampling and no time-resolved discharge data are available for

the sampling sites. We used annual mean discharge data, which were derived from data-driven regionalization of discharges from 125 gauging station from the period of 1979-1998 for the entire fluvial network. Application of discharge-weighted averaging was therefore not possible.

- Ln 128. Name the program used for statistics

Reply: All statistical analyses were performed with R. We will add this information to the revised manuscript.

- Ln 145-148. Discuss variance of organic C. How about peaty areas?

Reply: There are no pronounced regional or temporal differences of organic carbon. The fraction of peatland in the study area is small (0.95 km2 ; 0.009% of the study area) and only seven of the investigated catchments contain peaty areas. As organic C was measured only in three of these catchments, an investigation of the influence of peat on organic C was not possible. We would add the information about the variance of organic C and the peatland in the study area in a revised version of the manuscript.

- Ln 152. Specify which value is meant: mean NPP or mean specific NPP?

Reply: Mean specific NPP is meant. We will clarify this.

- Ln 169. In Figure 3 some of the data points (mostly stream order 1 and one of stream order 2) scatter more. Please discuss reasons for these outliers.

Reply: Small streams of low stream order can be directly influenced by local peculiarities which can increase the scatter of the data points while larger streams represent more average conditions over larger spatial scales. The scattering points in Figure 3 belong, e.g. to ditches or outflows from ponds which might differ in their characteristics to other rivers and streams. Based on the available data, however, there are no particular properties of all the scattering points, which would justify special treatment: the catchments are completely included, pH values are in the range of 7.1-8.3, and no urban areas around these catchments

Ln 186. You talk about average fluxes, but what happens during floods/ extreme events? Do measurement intervals cover extreme events?

Reply: Since we do not have time-resolved discharge data we cannot account for extreme events. Moreover, no information are available if the governmental monitoring included sampling during floods. Given the stochastic nature and short duration, we expect that such samples are at least underrepresented. Since it has been observed that high-discharge events can make a disproportionally high contribution to annual mean carbon export from catchments, we consider our estimates as a lower bound – in accordance with other uncertainty estimates, see below. We would add this information to the discussion in the revised manuscript.

- Ln 205. ": : :wetlands covering up to 16 % of the land surface area." Add a reference.

Reply: The 16% was taken from (Richey et al., 2002). However, in the revised version we would rather refer to a fraction of 14%, which was estimated in the study of (Abril et al., 2013). This reference is cited earlier in that sentence and will be moved to the end of the sentence in our revisions.

- Ln 226. Expected for temperate zones? In dry regions such as deserts this can be different.

Reply: We would change the sentence as follows: As expected for temperate zones, large streams and rivers with large surface area have larger catchments.

- Ln 235. Discuss "uncertainty of the various estimates"

Reply: For a comparable methodological approach, Butman and Raymond (2011) estimated the uncertainty in the calculation of the aquatic carbon flux to be 33% (based on Monte Carlo simulation). Raymond et al. (2013) estimated uncertainties from comparisons of estimates obtained using similar approaches as we with direct measurements of $CO_2$ concentration on streams. For a density of sampling locations of 0.02 sites per km2 (corresponding to our study) they derived an uncertainty of 30%. In addition to

errors associated with sampling and interpolation, our estimates are subject to a number of systematic errors. The neglect of carbon burial in sediments, carbon export and evasion as methane and unresolved flood events can be expected to result in an underestimation of the carbon exported from the catchments in our study. We will discuss these uncertainties at greater detail in the revised manuscript.

- Ln 236. Name potential controlling factors

Reply: Here we refer to the potentially controlling factors listed in Table 3, including catchment NPP, fractional water coverage as well as size and climatic zone of the study area. In the revised manuscript, we would list these factors in the text.

- Ln 244-247. How do you know that? In regions with corresponding geology also weathering of C-bearing minerals can be a large source of stream DIC. Respiration in soils is more likely the dominant DIC source in catchments that lack carbonate rocks. Is that true for catchments in Rhineland-Palatinate? Can you give an example for cases with predominance of aquatic respiration? I would expect predominance of aquatic respiration in warmer climates where large DOC concentrations prevail.

Reply: We agree. 16% of the study area contain carbonate bedrock and the observed association between DIC and the proportion of carbonate containing bedrock in the catchment ($R^2$=0.33, p<0.001) indicates, that weathering can be one additional source of DIC in the streams. We will add these results and revise the discussion accordingly. We will mention that both mineral weathering and soil respiration contribute to DIC in stream water and discuss the relative contributions of both sources observed in other studies (Hotchkiss et al., 2015;Lauerwald et al., 2013;Humborg et al., 2010;Jones et al., 2003). Examples for cases with predominance of aquatic respiration can not only be found in the tropics, but also in the boreal zone and in peat-draining streams. We would refer to (Duarte and Prairie, 2005;Jonsson et al., 2007;Lynch et al., 2010;Richey et al., 2002) as examples. (Hotchkiss et al., 2015) observed an increased $CO_2$ emissions from internal production for increasing stream size.

- Ln 249. How is the range of discharge? The study by Hotchkiss et al. covers values from 0.0001 to 10,000 m3 s-1. Can the lower range in your study be the reason that you do not observe findings in Hotchkiss et al.?

Reply: The range of discharge in our study is 0.003 - 12.2 m3 s-1 which is indeed a lower range compared to the study by Hotchkiss et al. We would add this in the discussion.

- Ln 252. Does "small number of observations" relate to this study?

Reply: No, at this point we refer to the meta-analysis presented in Table 3. We will make this clearer in the revised manuscript.

- Ln 255-257. This section summarizes the paper well but it could go further. It might be speculative but can you say what these other, poorly explored processes could be?

Reply: We would speculate, that hydrology plays a major role for C-cycling at larger scale. Precipitation controls not only terrestrial NPP but also, drainage density, export of OM from land to water and retention time of OM in soil and in surface waters respectively. Since our hydrological data base is rather weak (annual discharge only, no precipitation), we think that these speculations would be not well supported by the presented results.

- Ln 267. I think it is preferable to provide data as supplement material.

Reply: We will provide information about the investigated streams (stream order, water surface area, discharge, pH), catchment size and catchment NPP, DIC, DOC and TOC, pCO2 and the seasonality of pCO2, gas exchange velocity and total stream evasion as supplemental material.

- Table 1 and Table 2 would be more informative if you could add ranges. Please also add calculated gas transfer velocity values to Table 2.

Reply: We will add the information to the tables.

References:

Abril, G., Martinez, J.-M., Artigas, L. F., Moreira-Turcq, P., Benedetti, M. F., Vidal, L., Meziane, T., Kim, J.-H., Bernardes, M. C., Savoye, N., Deborde, J., Souza, E. L., Alberic, P., Landim de Souza, M. F., and Roland, F.: Amazon River carbon dioxide outgassing fuelled by wetlands, Nature, 505, 395-398, 10.1038/nature12797, 2013.

Abril, G., Bouillon, S., Darchambeau, F., Teodoru, C. R., Marwick, T. R., Tamooh, F., Ochieng Omengo, F., Geeraert, N., Deirmendjian, L., Polsenaere, P., and Borges, A. V.: Technical Note: Large overestimation of pCO2 calculated from pH and alkalinity in acidic, organic-rich freshwaters, Biogeosciences, 12, 67-78, 10.5194/bg-12-67-2015, 2015.

Butman, D., and Raymond, P. A.: Significant efflux of carbon dioxide from streams and rivers in the United States, Nature Geosci., 4, 839-842, 2011.

Duarte, C. M., and Prairie, Y. T.: Prevalence of Heterotrophy and Atmospheric CO2 Emissions from Aquatic Ecosystems, Ecosystems, 8, 862-870, 10.1007/s10021-005-0177-4, 2005.

Hotchkiss, E. R., Hall Jr, R. O., Sponseller, R. A., Butman, D., Klaminder, J., Laudon, H., Rosvall, M., and Karlsson, J.: Sources of and processes controlling CO2 emissions change with the size of streams and rivers, Nature Geosci., 8, 696-699, 10.1038/ngeo2507, 2015.

Humborg, C., Mörth, C.-M., Sundbom, M., Borg, H., Blenckner, T., Giesler, R., and Ittekkot, V.: CO2 supersaturation along the aquatic conduit in Swedish watersheds as constrained by terrestrial respiration, aquatic respiration and weathering, Global Change Biol., 16, 1966-1978, 10.1111/j.1365-2486.2009.02092.x, 2010.

Jones, J. B., Jr., Stanley, E. H., and Mulholland, P. J.: Long-term decline in carbon dioxide supersaturation in rivers across the contiguous United States, Geophys. Res. Lett., 30, 1495, 10.1029/2003gl017056, 2003.

Jonsson, A., Algesten, G., Bergström, A. K., Bishop, K., Sobek, S., Tranvik, L. J., and Jansson, M.: Integrating aquatic carbon fluxes in a boreal catchment carbon budget, J. Hydrol., 334, 141-150, 10.1016/j.jhydrol.2006.10.003, 2007.

Lauerwald, R., Hartmann, J., Moosdorf, N., Kempe, S., and Raymond, P. A.: What controls the spatial patterns of the riverine carbonate system? - A case study for North America, Chemical Geology, 337–338, 114-127, 10.1016/j.chemgeo.2012.11.011, 2013.

Lynch, J. K., Beatty, C. M., Seidel, M. P., Jungst, L. J., and DeGrandpre, M. D.: Controls of riverine CO2 over an annual cycle determined using direct, high temporal resolution pCO2 measurements, J. Geophys. Res., 115, G03016, 10.1029/2009jg001132, 2010.

Raymond, P. A., Hartmann, J., Lauerwald, R., Sobek, S., McDonald, C., Hoover, M., Butman, D., Striegl, R., Mayorga, E., Humborg, C., Kortelainen, P., Durr, H., Meybeck, M., Ciais, P., and Guth, P.: Global carbon dioxide emissions from inland waters, Nature, 503, 355-359, 10.1038/nature12760, 2013.

Richey, J. E., Melack, J. M., Aufdenkampe, A. K., Ballester, V. M., and Hess, L. L.: Outgassing from Amazonian rivers and wetlands as a large tropical source of atmospheric CO2, Nature, 416, 617-620, 10.1038/416617a, 2002.

Stanley, E. H., Casson, N. J., Christel, S. T., Crawford, J. T., Loken, L. C., and Oliver, S. K.: The ecology of methane in streams and rivers: patterns, controls, and global significance, Ecol. Monographs, 86, 146-171, 10.1890/15-1027, 2016.

---

## Author Comment (AC2) · 28 Apr 2017

Anonymous Referee #2:

Accurate estimation of aquatic carbon export is essential to understand the role of natural ecosystems and geochemical processes in global carbon cycles in the context of climate change and increasing anthropogenic activities. In this manuscript, the authors integrate the analysis of downstream export of riverine carbon and CO2 evasion to the atmosphere from more than 200 local catchments of variable sizes in temperate Europe along with the model estimation of ecosystem production. Based on this large dataset, the authors try to establish a carbon budget in a local scale and discuss the ecologic factors controlling the aquatic carbon export. Overall, the integration of the large dataset of riverine carbon concentrations spanning over last several decades is

technically sound and strengthens the arguments in the manuscript.

We would like to thank the reviewer for her/his positive evaluation and the very helpful comments and suggestions. Below we reply on each specific comment.

- My biggest concern arises from the estimation of the downstream export of riverine carbon. The riverine carbon concentrations adopted in this investigation were obtained during 1977-2011, which is significantly longer than NPP of 2000-2013. Investigations have already showed a decadal increasing DIC export in boreal and subtropical rivers due to the climate change and anthropogenic activities (Walvoord, M. A., and R. G. Striegl, 2007, Increased groundwater to stream discharge from permafrost thawing in the Yukon River basin: Potential impacts on lateral export of carbon and nitrogen, Geophys. Res. Lett., 34, L12402, doi:10.1029/2007GL030216; Raymond, P.A., Oh, N.-H., Turner, R.E., Broussard, W., 2008. Anthropogenically enhanced fluxes of water and carbon from the Mississippi River. Nature 451, 449-452). Therefore, I would suggest using the environment monitoring dataset during the last 10 years or so, which is consistemt with NPP estimation, to estimate the riverine carbon export.

Reply: Using only the monitoring data during the last 10 years would reduce the number of valid samples from currently 8020 to 5070. We compared DIC measured in the time periods 1977-1999 and 2000-2011 for all Strahler orders. The DIC did not change significantly. We would provide this information in a revised version of the manuscript and also point towards the trends observed in other regions, as mentioned by the reviewer.

- Secondly, it seems that the data points for the flux estimation is sparse as indicated in the section 2.2 (see Page 3 Line 83-86: ": : :: : :at least one measurement was available for each season: : :: : :"). Therefore, I will be happy to see the error or uncertainty analysis of the flux estimation with the method using the mean concentration and total river discharge (see Page 4 Line 94-95). Moreover, a comparison with other flux estimation methods, such as the one using flow-weighted mean concentration and

discharge, the one based on the regression of instantaneous flux and discharge, and other methods (see Warnken, K.W., Santschi, P.H., 2004. Biogeochemical behavior of organic carbon in the Trinity River downstream of a large reservoir lake in Texas, USA. Sci. Total Environ. 329, 131-144), will be helpful to validate the flux estimation.

Reply: Unfortunately, the only available discharge data are annual mean values derived using data-driven regionalization of discharges from 125 gauging station from the period of 1979-1998. Time-resolved discharge measurements or data for the sampling times and sites, which could be used for flow-weighted estimates of DIC export and $CO_2$ evasion, are not available. For an uncertainty analysis, we would use the more extensive analysis of Raymond et al. (2013), where uncertainties were derived based on comparisons of estimates obtained using similar approaches as we used with direct measurements of $CO_2$ concentration. For a density of sampling locations of 0.02 sites per km2 they derived an uncertainty of 30%. Similarly, Butman and Raymond (2011) estimated uncertainties of overall flux estimates of 33%, based on Monte Carlo simulation of similar data for hydrographic units in the United States. In addition to errors associated with sampling and interpolation, our estimates are subject to a number of systematic errors. The neglect of carbon burial in sediments, carbon export and evasion as methane and under-sampling of high-discharge events, probably result in an underestimation of the carbon exported from the catchments in our study. We will discuss these uncertainties at greater detail in the revised manuscript.

- What do you mean "interpolating pCO2 for all river segments without direct measurement" (Page 4 Line 95-97)? Please clarify in the text.

Reply: The explanation is provided in the following sentence (Line 97-98): For this, the mean concentrations were averaged by stream order and assigned to all stream segments of the river network (Butman and Raymond, 2011). To clarify this, we would join the two sentences in a revised manuscript.

- For DOC, there are 64 observations (Table 1) in 54 sampling sites (Page 3 Line

90- 91). On average, there are less than 2 observations in each site. Usually, DOC concentrations in rivers could vary seasonally with river discharge by couples of times. Therefore, the representativeness of the single DOC data in each catchment remains a critical question which may induce the great deviation of DOC flux estimation from the real value. Before resolving this issue, the statements that DOC load only made up 4% total carbon load (Page 5 Line 146-148) and that the error would be comparably small when neglecting the DOC term (Page 6 Line 159-162) seem arbitrary.

Reply: 54 is a mistake in writing, which we will correct. There are seasonally averaged DOC observations at 64 sampling sites; all values are averaged over several measurements covering all seasons.

- The authors extensively discuss the aquatic carbon export/NPP ratio in the manuscript (See Table 3 and text in Section 4.1s). They state in the manuscript:" By combining CO2 evasion and downstream C-export by stream discharge, we estimated that 2.7 % of terrestrial NPP (13.9 g C m2 yr-1) are exported from the catchments by streams and rivers, in which both evasion and discharge contributed equally to this flux (Page 7 Line 193-195)". Then they compare their results with some other studies of catchment ecosystems (see text in Section 4.2). However, what I understand is riverine DIC export flux is closely related to the weathering regimes and intensity in catchments (See Cai, W.-J., Guo, X., Chen, C.-T.A., Dai, M., Zhang, L., Zhai, W., Lohrenz, S.E., Yin, K., Harrison, P.J., Wang, Y., 2008. A comparative overview of weathering intensity and HCO3- flux in the world's major rivers with emphasis on the Changjiang, Huanghe, Zhujiang (Pearl) and Mississippi Rivers. Continental Shelf Research 28, 1538-1549; and Raymond, P.A., Bauer, J.E., Caraco, N.F., Cole, J.J., Longworth, B., Petsch, S.T., 2004. Controls on the variability of organic matter and dissolved inorganic carbon ages in northeast US rivers. Marine Chemistry 92, 353-366) although NPP could contribute part of DIC export flux through the respiration of DOM. Therefore, the aquatic carbon export/NPP ratio would be expected to be larger than the real contribution of NPP.

Reply: Our analysis of aquatic C-export in relation to NPP was inspired by studies

where correlations between aquatic C export and terrestrial NPP or NEP have been observed at different spatial scales and different landscapes (e.g. Butman et al., 2015;Maberly et al., 2013; and other studies listed in Table 3). The reason for a lack of correlation in our study could be related to weathering, as pointed out by the reviewer. 16% of the study area contain carbonate bedrock. In our study area, the DIC concentration in the water increased with the proportion of carbonate containing bedrock in the catchment ($R^2=0.33$, $p<0.001$). In the discussion of the revised manuscript, we would list this as an uncertainty and add estimates of the contribution of weathering found in other studies.

References:

Butman, D., and Raymond, P. A.: Significant efflux of carbon dioxide from streams and rivers in the United States, Nature Geosci., 4, 839-842, 2011.

Butman, D., Stackpoole, S., Stets, E., McDonald, C. P., Clow, D. W., and Striegl, R. G.: Aquatic carbon cycling in the conterminous United States and implications for terrestrial carbon accounting, Proceedings of the National Academy of Sciences, 10.1073/pnas.1512651112, 2015.

Maberly, S. C., Barker, P. A., Stott, A. W., and De Ville, M. M.: Catchment productivity controls CO2 emissions from lakes, Nat. Clim. Chang., 3, 391-394, 10.1038/nclimate1748, 2013.

Raymond, P. A., Hartmann, J., Lauerwald, R., Sobek, S., McDonald, C., Hoover, M., Butman, D., Striegl, R., Mayorga, E., Humborg, C., Kortelainen, P., Durr, H., Meybeck, M., Ciais, P., and Guth, P.: Global carbon dioxide emissions from inland waters, Nature, 503, 355-359, 10.1038/nature12760, 2013.

---

## Author Comment (AC4) · 28 Apr 2017

Anonymous Referee #3:

Katrin Magin and colleagues presented a synthesis of >200 catchments examining the relationships between lateral carbon export and CO2 emissions and terrestrial net primary production (NPP) in southwest Germany. Inland waters have recently been recognized as important components in the global carbon cycle. While widespread studies have been conducted worldwide, most of these studies are based on individual catchments and a synthesis involving multiple catchments remains lacking. This manuscript is well-organized and quite timely, and will provide insights into the understanding of catchment carbon cycle (or budget) at regional scales.

We would like to thank the reviewer for her/his positive evaluation and the very helpful

comments and suggestions. Below we reply on each specific comment.

- My first major concern after reading this manuscript is the carbon storage term which has not yet been considered when the authors evaluated catchment-scale carbon budget. Caron burial associated with soil erosion and sediment deposition within catchments is a quite important component in carbon budget assessments (e.g., Smith et al., 2001). If the traditional sediment delivery ratio of 10% is assumed (Harden et al.1999), 90% of the eroded POC from land may have been stored somewhere within the catchment and partly exposed to decomposition (thus evasion to the atmosphere). This missing term may affect the redistribution of carbon (downstream discharge vs. CO2 evasion) as well as the amount of total carbon input from land. Incorporating this term will thus refine the budget result.

Reply: See our response to your specific comment below.

- My second concern is the estimation of CO2 evasion. What are the resulting k600 values? Are they comparable to those based on field direct measurements (e.g., floating chamber or eddy covariance)? Estimation of the total areal extent of water surface by means of the parameters derived from USA catchments is probably problematic (see my specific comment below). In addition, can the available dataset suggest any seasonal variability in CO2 evasion?

Reply: The k600 values in our study range from 2.0 m d-1 to 20.6 m d-1 with a mean of 6.0±3.3 m d-1. These transfer velocities are comparable to k600 values based on direct field measurements by floating chambers from small headwater streams in Alaska (Crawford et al., 2013) and also to some short chamber deployments within the study area (Lorke et al., 2015). The pCO2 is higher in summer (mean±sd: 2780±2098 ppm) and autumn (mean±sd: 2848±2019 ppm) than in winter (mean±sd: 2287±1716 ppm) and spring (mean±sd: 2172±2343 ppm). In contrast, the relationship of catchment NPP and CO2 evasion is not influenced by the season. We will add the k600 values in Table 2 and discuss the seasonal variability of pCO2 in a revised version of the

manuscript.

- Line 19: please clarify 'catchment-specific total export rate'. Is it the normalized carbon export by catchment area?

Reply: Yes, the catchment-specific carbon export rate refers to the carbon export per catchment area. We will clarify this in a revised version of the manuscript.

- Line 29-30: the latest $CO_2$ evasion from global rivers and streams is 0.65 Pg C/yr by Lauerwald et al., (2015).

Reply: We will include the reference in this section.

- Line 50: remove 'differ'.

- Line 71: the reference 'Strahler, 1957' should move to line 56.

- Line 77: remaining →retained

Reply: We will apply these corrections.

- Line 81-83. What's the data quality and what kinds of standards for water sampling and processing were used? Estimating $pCO_2$ from alkalinity and pH has been criticized for causing biases due to noncarbonate impacts (Abril et al., 2015). An uncertainty analysis should be provided here. I also suggest to provide the range of pH and alkalinity, possibly into Table 1.

Reply: The data are governmental monitoring data which are acquired according to DIN EN ISO norms ((DIN EN ISO 10523:2012-04;DIN EN ISO 9963-1:1996-02;DIN EN ISO 9963-2:1996-02)). The range of pH values of the investigated waters was $6.2 - 8.97$ with a mean of $7.73\pm0.42$ (mean$\pm$sd). The range of alkalinity is $0.08 - 9.88$ mmol L-1 with a mean of $2.75\pm2.12$ mmol L-1 (mean$\pm$sd). We would add pH values and alkalinity in Table 1. According to Abril et al. (2015), high uncertainties of $pCO_2$ estimates from pH and alkalinity measurements occur at pH values <7, while the median and mean relative errors were 1% and 15%, respectively for pH>7. Only

7% of the pH values in our study were <7. We would add a discussion of the expected uncertainties to a revised manuscript.

- Line 95-97: how was the site-specific pCO2 interpolated to the upstream catchments? And which interpolation technique was used?

Reply: The explanation is provided in the following sentence (Line 97-98): For this, the mean concentrations were averaged by stream order and assigned to all stream segments of the river network (Butman and Raymond, 2011). To clarify this, we would join the two sentences in a revised manuscript.

- Line 102-103: These arbitrary parameters derived from American rivers may not necessarily be representative of German rivers. See Leopold and Maddock (1953).

Reply: The coefficients we used were derived from various data sets obtained in North America, but have been applied also in global studies before, e.g. Raymond et al. (2013). Unfortunately, we are not aware of a comparably extensive data set of hydraulic geometry data derived for European rivers. A comparison of hydraulic geometry co-efficients derived from various data sets, including data from England, Australia and New Zealand, is presented in Butman and Raymond (2011), who estimated that the error associated with uncertainties of hydraulic geometry coefficients is rather small, compared to uncertainties derived for C-fluxes. We will add these information to an extended discussion of uncertainties in the revised manuscript.

- Line 105: Is a resolution of 10 m enough to estimate channel slope changes?

Reply: Zhang and Montgomery (1994) investigated the effect of digital elevation model (DEM) resolution on slope calculation and performance in hydrological models for spatial resolutions between 2 and 90 m. They found that while a 10-m grid is a significant improvement over 30 m or coarser grid sizes, finer grid sizes provide relatively little additional resolution. Thus a 10-m grid size presents a reasonable compromise between increasing spatial resolution and data handling requirements for modeling surface processes in many landscapes. We will justify the choice of DEM resolution in the revised manuscript. It should be noted, that similar studies have derived slope information from coarser DEM resolution, e.g. SRTM 90m Digital Elevation Data (Lauerwald et al., 2013), GMTED2010 with >250 m resolution (Raymond et al., 2013), NHDPlus with 30 m ground resolution (Butman et al., 2015).

- Line 125-126: Because the mean NPP for the period 2000-2013 is used here while the $pCO_2$ data is for the period 1970-2011, it is better to explicitly indicate the distribution frequency of $pCO_2$ data over the study period. For example, if the most of the $pCO_2$ data were for the period 1970-1980, then using the NPP for 2000-2013 would be problematic.

Reply: The sampling frequency was increasing. There are nearly twice as much data from 2000-2011 than from 1977-1999. A comparison between DIC data from both sampling periods revealed no significant differences. We would add the sampling frequency distribution in a revised version of the manuscript.

- Line 135: Based on the given definition, the 'drainage rate' term should be 'runoff depth' in a formal way.

Reply: We would correct the term in our revisions.

- Line 146: Please quantify 'only a small fraction'.

Reply: On average 8.6% of the TOC consist of POC. The highest percentage of POC found in a catchment is 28.2%.

- Line 158: For the total C input, how about the POC term and the carbon storage term? See my major comment.

Reply: POC as suspended load in the rivers was estimated along with DOC and was only 8.6% of the TOC load (0.8 % of the total C-load) at the sampling sites. We agree with the reviewer that storage can make a significant contribution to the catchment-scale C balance. Estimates vary between 22% at a global scale (Aufdenkampe et

al., 2011), 14 % for the Conterminous U.S. (Butman et al., 2015) and 39% for the Yellow River network (Ran et al., 2015). However, C storage in aquatic systems occurs mainly in lakes and reservoirs, which are virtually absent in the catchments studied here. Therefore we consider the bias caused by neglecting storage to be comparable in magnitude to remaining uncertainties (30%). We would add a more detailed discussion of the storage term and the associated uncertainty as part of a general uncertainty analysis (see comments above) in a revised discussion section. The neglect of storage, and potential high C loads during extreme discharge events, suggest that C-export from catchments estimated in the present study provides a lower bound of the aquatic C flux.

- Line 220-223: Are there peatlands within the studied catchments?

Reply: The fraction of peatlands in the area in really small (0.009%) and is restricted to 7 of the investigated catchments. For only 3 of these catchments DOC measurements were available and no influence of the peatland on the DOC was observable. We would add the information about the peatland in the study area in a revised version of the manuscript.

- Line 229-230: Is the absence of the carbon yield and NPP correlation due to failure to measure pCO2 during flooding periods? The short-duration carbon export during flooding events usually accounts for disproportionately a large share of the annual total carbon export.

Reply: Since we do not have time-resolved discharge data we cannot account for extreme events. Moreover, no information are available if the governmental monitoring included sampling during floods. Given the stochastic nature and short duration, we expect that such samples are at least underrepresented. Since it has been observed that high-discharge events can make a disproportionally high contribution to annual mean carbon export from catchments, we consider our estimates as a lower bound – in accordance with other uncertainty estimates, see below. We would add this information to the discussion in the revised manuscript.

- Line 238: please clarify the 'surface area'. The global surface area?

Reply: No, this surface area refers to regions in the 2 studies cited in that sentence. We will clarify this in our revisions.

- Line 244-247: Could it also be because of chemical weathering and groundwater inputs? Rock weathering in carbonate-dominated catchments can be a significant contributor to DIC. I would suggest the authors to make a brief introduction about the lithology and mineralogy in the study area section (2.1).

Reply: 16% of the area investigated in our study contain calcareous bedrock. The DIC concentration in the water increased with the proportion of carbonate bedrock in the catchments (R2=0.33, p<0.001). We would add the information about the bedrock in the study area section and include the influence of chemical weathering in our discussion.

- Line 262: please summarize the study and make a short conclusion.

Reply: We would add the following conclusion to a revised version of the manuscript: Our analysis of the carbon budget in a temperate stream network on regional scale revealed a relationship of aquatic carbon export and terrestrial NPP. On average 2.7% of the terrestrial NPP were exported from the catchments by rivers and streams with CO2 evasion and downstream transport contributing equally to the export. A comparison of our regional scale study with other studies from different scales and landscapes showed a relatively narrow range of variability of carbon export per catchment area. Future research is needed to understand the processes that control the aquatic-terrestrial coupling and the role of inland waters in regional carbon cycling.

- Figure 2. It seems the top 2(?) data points far away from the majority are outliers. Please check and make the regression again, if necessary.

Reply: The apparent outlier is only 1 data point. It does not influence the regression.

- Table 1. pH could also be tabulated here. Is there any trend in pH from SO1 to SO4?

Reply: We would include pH in Table 1. There is no trend in pH from SO1 to SO4 but the variability of pH (e.g. standard deviation) is decreasing with increasing stream order.

References:

Abril, G., Bouillon, S., Darchambeau, F., Teodoru, C. R., Marwick, T. R., Tamooh, F., Ochieng Omengo, F., Geeraert, N., Deirmendjian, L., Polsenaere, P., and Borges, A. V.: Technical Note: Large overestimation of pCO2 calculated from pH and alkalinity in acidic, organic-rich freshwaters, Biogeosciences, 12, 67-78, 10.5194/bg-12-67-2015, 2015.

Aufdenkampe, A. K., Mayorga, E., Raymond, P. A., Melack, J. M., Doney, S. C., Alin, S. R., Aalto, R. E., and Yoo, K.: Riverine coupling of biogeochemical cycles between land, oceans, and atmosphere, Front. Ecol. Environ., 9, 53-60, 10.1890/100014, 2011.

Butman, D., and Raymond, P. A.: Significant efflux of carbon dioxide from streams and rivers in the United States, Nature Geosci., 4, 839-842, 2011.

Butman, D., Stackpoole, S., Stets, E., McDonald, C. P., Clow, D. W., and Striegl, R. G.: Aquatic carbon cycling in the conterminous United States and implications for terrestrial carbon accounting, Proceedings of the National Academy of Sciences, 10.1073/pnas.1512651112, 2015.

Crawford, J. T., Striegl, R. G., Wickland, K. P., Dornblaser, M. M., and Stanley, E. H.: Emissions of carbon dioxide and methane from a headwater stream network of interior Alaska, Journal of Geophysical Research: Biogeosciences, 118, 482-494, 2013.

DIN EN ISO 9963-1:1996-02: Wasserbeschaffenheit - Bestimmung der Alkalinität - Teil 1: Bestimmung der gesamten und der zusammengesetzten Alkalinität (ISO 9963-1:1994); German version EN ISO 9963-1:1995.

DIN EN ISO 9963-2:1996-02: Wasserbeschaffenheit - Bestimmung der Alkalinität - Teil 2: Bestimmung der Carbonatalkalinität (ISO 9963-2:1994); German version EN

ISO 9963-2:1995.

DIN EN ISO 10523:2012-04: Wasserbeschaffenheit - Bestimmung des pH-Werts (ISO 10523:2008); German version EN ISO 10523:2012.

Lauerwald, R., Hartmann, J., Moosdorf, N., Kempe, S., and Raymond, P. A.: What controls the spatial patterns of the riverine carbonate system? - A case study for North America, Chemical Geology, 337–338, 114-127, 10.1016/j.chemgeo.2012.11.011, 2013.

Lorke, A., Bodmer, P., Noss, C., Alshboul, Z., Koschorreck, M., Somlai-Haase, C., Bastviken, D., Flury, S., McGinnis, D. F., Maeck, A., Müller, D., and Premke, K.: Technical note: drifting versus anchored flux chambers for measuring greenhouse gas emissions from running waters, Biogeosciences, 12, 7013-7024, 10.5194/bg-12-7013-2015, 2015.

Ran, L., Lu, X. X., Yang, H., Li, L., Yu, R., Sun, H., and Han, J.: CO2 outgassing from the Yellow River network and its implications for riverine carbon cycle, J. Geophys. Res.-Biogeo., 120, 1334-1347, 10.1002/2015JG002982, 2015.

Raymond, P. A., Hartmann, J., Lauerwald, R., Sobek, S., McDonald, C., Hoover, M., Butman, D., Striegl, R., Mayorga, E., Humborg, C., Kortelainen, P., Durr, H., Meybeck, M., Ciais, P., and Guth, P.: Global carbon dioxide emissions from inland waters, Nature, 503, 355-359, 10.1038/nature12760, 2013. Zhang, W., and Montgomery, D. R.: Digital elevation model grid size, landscape representation, and hydrologic simulations, Water Resour. Res., 30, 1019-1028, 10.1029/93WR03553, 1994.
* * *

---

## Author Response (AR1)

**(1) Comments from referees**

Anonymous Referee #1

General Comments

Rivers and streams are an important link in the global C cycle and C export via aquatic systems has repeatedly been concluded to make a significant proportion of catchment C budgets at different spatial scales and in different climate zones. This is an interesting paper that will make a good contribution to the understanding of regional-scale carbon export via streams with stream order 1-4 in the temperate zone. The authors observed a narrow range of variability of C export per catchment area and conclude that other processes than water surface area or location of mineralization of terrestrial derived C control the aquatic-terrestrial coupling and the role of inland waters in regional C cycling. However, the final version of the paper would benefit from more details about lateral C export calculations. It is not clear if extreme runoff events are covered appropriately. The lack of extreme event data would of course lead to a much narrower range of variability of C export.

Specific Comments

Ln 20. Explain "catchment-specific"

Ln 53. Why is the fluvial C load dominated by DIC? Are there carbonates? Please also state why you neglected methane.

Ln 56. Strahler stream order?

Ln 61-66. What about the geology? Is there C-containing bedrock in the catchments?

Ln 70. "15 800" do not separate numbers

Ln 71. Delete "order"

Ln 82. How are pH values of investigated waters? The pCO2 calculation with alkalinity was found to high uncertainties for low pH values (Abril et al. 2014).

Ln 89. How exactly did you aggregate annual means? Did you calculate a (discharge) weighted average?

Ln 128. Name the program used for statistics

Ln 145-148. Discuss variance of organic C. How about peaty areas?

Ln 152. Specify which value is meant: mean NPP or mean specific NPP?

Ln 169. In Figure 3 some of the data points (mostly stream order 1 and one of stream order 2) scatter more. Please discuss reasons for these outliers.

Ln 186. You talk about average fluxes, but what happens during floods/ extreme events? Do measurement intervals cover extreme events?

Ln 205. ": : :wetlands covering up to 16 % of the land surface area." Add a reference.

Ln 226. Expected for temperate zones? In dry regions such as deserts this can be different.

Ln 235. Discuss "uncertainty of the various estimates"

Ln 236. Name potential controlling factors

Ln 244-247. How do you know that? In regions with corresponding geology also weathering of C-bearing minerals can be a large source of stream DIC. Respiration in soils is more likely the dominant DIC source in catchments that lack carbonate rocks. Is that true for catchments in Rhineland-Palatinate? Can you give an example for cases with predominance of aquatic respiration? I would expect predominance of aquatic respiration in warmer climates where large DOC concentrations prevail.

Ln 249. How is the range of discharge? The study by Hotchkiss et al. covers values from 0.0001 to 10,000 m3 s-1. Can the lower range in your study be the reason that you do not observe findings in Hotchkiss et al.?

Ln 252. Does "small number of observations" relate to this study?

Ln 255-257. This section summarizes the paper well but it could go further. It might be speculative but can you say what these other, poorly explored processes could be?

Ln 267. I think it is preferable to provide data as supplement material.

Table 1 and Table 2 would be more informative if you could add ranges. Please also add calculated gas transfer velocity values to Table 2.

References:

Abril, G., S. Bouillon, F. Darchambeau, C. R. Teodoru, T. R. Marwick, F. Tamooh, F. O. Omengo, N. Geeraert, L. Deirmendjian, P. Polsenaere, and A. V. Borges (2015), Technical Note: Large overestimation of pCO2 calculated from pH and alkalinity in acidic, organic-rich freshwaters, Biogeosciences, 12(1), 67-78, doi:10.5194/Bg-12-67-2015.

Anonymous Referee #2

Accurate estimation of aquatic carbon export is essential to understand the role of natural ecosystems and geochemical processes in global carbon cycles in the context of climate change and increasing anthropogenic activities. In this manuscript, the authors integrate the analysis of downstream export of riverine carbon and CO2 evasion to the atmosphere from more than 200 local catchments of variable sizes in temperate Europe along with the model estimation of ecosystem production. Based on this large dataset, the authors try to establish a carbon budget in a local scale and discuss the ecologic factors controlling the aquatic carbon export. Overall, the integration of the large dataset of riverine carbon concentrations spanning over last several decades is technically sound and strengthens the arguments in the manuscript.

My biggest concern arises from the estimation of the downstream export of riverine carbon. The riverine carbon concentrations adopted in this investigation were obtained during 1977-2011, which is significantly longer than NPP of 2000-2013. Investigations have already showed a decadal increasing DIC export in boreal and subtropical rivers due to the climate change and anthropogenic activities (Walvoord, M. A., and R. G. Striegl, 2007, Increased groundwater to stream discharge from permafrost thawing in the Yukon River basin: Potential impacts on lateral export of carbon and nitrogen, Geophys. Res. Lett., 34, L12402, doi:10.1029/2007GL030216; Raymond, P.A., Oh,

N.-H., Turner, R.E., Broussard, W., 2008. Anthropogenically enhanced fluxes of water and carbon from the Mississippi River. Nature 451, 449-452). Therefore, I would suggest using the environment monitoring dataset during the last 10 years or so, which is consistemt with NPP estimation, to estimate the riverine carbon export.

Secondly, it seems that the data points for the flux estimation is sparse as indicated in the section 2.2 (see Page 3 Line 83-86: ": : :: : :at least one measurement was available for each season: : :: : :"). Therefore, I will be happy to see the error or uncertainty analysis of the flux estimation with the method using the mean concentration and total river discharge (see Page 4 Line 94-95). Moreover, a comparison with other flux estimation methods, such as the one using flow-weighted mean concentration and discharge, the one based on the regression of instantaneous flux and discharge, and other methods (see Warnken, K.W., Santschi, P.H., 2004. Biogeochemical behavior of organic carbon in the Trinity River downstream of a large reservoir lake in Texas, USA. Sci. Total Environ. 329, 131-144), will be helpful to validate the flux estimation.

What do you mean "interpolating pCO2 for all river segments without direct measurement" (Page 4 Line 95-97)? Please clarify in the text.

For DOC, there are 64 observations (Table 1) in 54 sampling sites (Page 3 Line 90-91). On average, there are less than 2 observations in each site. Usually, DOC concentrations in rivers could vary seasonally with river discharge by couples of times. Therefore, the representativeness of the single DOC data in each catchment remains a critical question which may induce the great deviation of DOC flux estimation from the real value. Before resolving this issue, the statements that DOC load only made up 4% total carbon load (Page 5 Line 146-148) and that the error would be comparably small when neglecting the DOC term (Page 6 Line 159-162) seem arbitrary.

The authors extensively discuss the aquatic carbon export/NPP ratio in the manuscript (See Table 3 and text in Section 4.1s). They state in the manuscript:" By combining CO2 evasion and downstream C-export by stream discharge, we estimated that 2.7 % of terrestrial NPP (13.9 g C m2 yr-1) are exported from the catchments by streams and rivers, in which both evasion and discharge contributed equally to this flux (Page 7 Line 193-195)". Then they compare their results with some other studies of catchment ecosystems (see text in Section 4.2). However, what I understand is riverine DIC export flux is closely related to the weathering regimes and intensity in catchments (See Cai, W.-J., Guo, X., Chen, C.-T.A., Dai, M., Zhang, L., Zhai, W., Lohrenz, S.E., Yin, K., Harrison, P.J., Wang, Y., 2008. A comparative overview of weathering intensity and HCO3- flux in the world's major rivers with emphasis on the Changjiang, Huanghe, Zhujiang (Pearl) and Mississippi Rivers. Continental Shelf Research 28, 1538-1549; and Raymond, P.A., Bauer, J.E., Caraco, N.F., Cole, J.J., Longworth, B., Petsch, S.T., 2004. Controls on the variability of organic matter and dissolved inorganic carbon ages in northeast US rivers. Marine Chemistry 92, 353-366) although NPP could contribute part of DIC export flux through the respiration of DOM. Therefore, the aquatic carbon export/NPP ratio would be expected to be larger than the real contribution of NPP.

Anonymous Referee #1

Please check equation 1. The exponents and coefficients between width and depth are switched. In "Stream
Hydraulics" in Raymond et al. (2012) this can be checked. I think the correct formulas would be w=12.88*Q^0.42
and d=0.4*Q^0.29. This corresponds to width=c*Q^d and depth=a*Q^b. The coefficients and exponents of your
equation lead to depth»width, which seems not realistic.

Anonymous Referee #3
General comments:
Katrin Magin and colleagues presented a synthesis of >200 catchments examining the relationships between lateral
carbon export and CO2 emissions and terrestrial net primary production (NPP) in southwest Germany. Inland waters
have recently been recognized as important components in the global carbon cycle. While widespread studies have
been conducted worldwide, most of these studies are based on individual catchments and a synthesis involving
multiple catchments remains lacking. This manuscript is well-organized and quite timely, and will provide insights
into the understanding of catchment carbon cycle (or budget) at regional scales.
My first major concern after reading this manuscript is the carbon storage term which has not yet been considered
when the authors evaluated catchment-scale carbon budget. Caron burial associated with soil erosion and sediment
deposition within catchments is a quite important component in carbon budget assessments (e.g., Smith et al., 2001).
If the traditional sediment delivery ratio of 10% is assumed (Harden et al., 1999), 90% of the eroded POC from land
may have been stored somewhere within the catchment and partly exposed to decomposition (thus evasion to the
atmosphere). This missing term may affect the redistribution of carbon (downstream discharge vs. CO2 evasion) as
well as the amount of total carbon input from land. Incorporating this term will thus refine the budget result.
My second concern is the estimation of CO2 evasion. What are the resulting k600 values? Are they comparable to
those based on field direct measurements (e.g., floating chamber or eddy covariance)? Estimation of the total areal
extent of water surface by means of the parameters derived from USA catchments is probably problematic (see my
specific comment below). In addition, can the available dataset suggest any seasonal variability in CO2 evasion?
Specific comments:
Line 19: please clarify 'catchment-specific total export rate'. Is it the normalized carbon export by catchment area?
Line 29-30: the latest CO2 evasion from global rivers and streams is 0.65 Pg C/yr by Lauerwald et al., (2015).
Line 50: remove 'differ'.
Line 71: the reference 'Strahler, 1957' should move to line 56.
Line 77: remaining!retained
Line 81-83. What's the data quality and what kinds of standards for water sampling and processing were used?
Estimating pCO2 from alkalinity and pH has been criticized for causing biases due to noncarbonate impacts (Abril
et al., 2015). An uncertainty analysis should be provided here. I also suggest to provide the range of pH and
alkalinity, possibly into Table 1.

Line 95-97: how was the site-specific pCO2 interpolated to the upstream catchments? And which interpolation technique was used?

Line 102-103: These arbitrary parameters derived from American rivers may not necessarily be representative of German rivers. See Leopold and Maddock (1953).

Line 105: Is a resolution of 10m enough to estimate channel slope changes?

Line 125-126: Because the mean NPP for the period 2000-2013 is used here while the pCO2 data is for the period 1970-2011, it is better to explicitly indicate the distribution frequency of pCO2 data over the study period. For example, if the most of the pCO2 data were for the period 1970-1980, then using the NPP for 2000-2013 would be problematic.

Line 135: Based on the given definition, the 'drainage rate' term should be 'runoff depth' in a formal way.

Line 146: Please quantify 'only a small fraction'.

Line 158: For the total C input, how about the POC term and the carbon storage term? See my major comment.

Line 220-223: Are there peatlands within the studied catchments?

Line 229-230: Is the absence of the carbon yield and NPP correlation due to failure to measure pCO2 during flooding periods? The short-duration carbon export during flooding events usually accounts for disproportionately a large share of the annual total carbon export.

Line 238: please clarify the 'surface area'. The global surface area?

Line 244-247: Could it also be because of chemical weathering and groundwater inputs? Rock weathering in carbonate-dominated catchments can be a significant contributor to DIC. I would suggest the authors to make a brief introduction about the lithology and mineralogy in the study area section (2.1).

Line 262: please summarize the study and make a short conclusion.

Figure 2. It seems the top 2(?) data points far away from the majority are outliers. Please check and make the regression again, if necessary.

Table 1. pH could also be tabulated here. Is there any trend in pH from SO1 to SO4?

References

Abril, G., Bouillon, S., Darchambeau, F., Teodoru, C., Marwick, T., Tamooh, F., Omengo, F., Geeraert, N., Deirmendjian, L., Polsenaere, P., and Borges, A. V.: Technical Note: Large overestimation of pCO2 calculated from pH and alkalinity in acidic, organic-rich freshwaters, Biogeosciences, 12, 67-78, 2015.

Harden, J., Sharpe, J., Parton, W., Ojima, D., Fries, T., Huntington, T., and Dabney, S.: Dynamic replacement and loss of soil carbon on eroding cropland, Global Biogeochem. Cy., 13, 885–901, 1999.

Lauerwald, R., Laruelle, G. G., Hartmann, J., Ciais, P., and Regnier, P. A.: Spatial patterns in CO2 evasion from the global river network, Global Biogeochemical Cycles, 29, 534-554, 2015.

Leopold, L., and Maddock, T.: The hydraulic geometry of stream channels and some physiographic implications, USGS Professional Paper 252, USGS Professional Paper 252, 1953.

Smith, S. V., Renwick, W. H., Buddemeier, R. W., and Crossland, C. J.: Budgets of soil erosion and deposition for sediments and sedimentary organic carbon across the conterminous United States, Global Biogeochemical Cycles,

15, 697-707, 2001.

    **(2) Author's response**

    **Anonymous Referee #1**

Rivers and streams are an important link in the global C cycle and C export via aquatic systems
has repeatedly been concluded to make a significant proportion of catchment C budgets at
different spatial scales and in different climate zones. This is an interesting paper that will make
a good contribution to the understanding of regional-scale carbon export via streams with stream
order 1-4 in the temperate zone. The authors observed a narrow range of variability of C export
per catchment area and conclude that other processes than water surface area or location of
mineralization of terrestrial derived C control the aquatic-terrestrial coupling and the role of
inland waters in regional C cycling. However, the final version of the paper would benefit from
more details about lateral C export calculations. It is not clear if extreme runoff events are
covered appropriately. The lack of extreme event data would of course lead to a much narrower
range of variability of C export.

We would like to thank the reviewer for her/his positive evaluation and the very helpful
comments and suggestions. Below we reply on each specific comment.
- Ln 20. Explain "catchment-specific"
Reply: Catchment-specific carbon export refers to the carbon export per catchment area. We
explained this in the revised version of the manuscript.
- Ln 53. Why is the fluvial C load dominated by DIC? Are there carbonates? Please also
state why you neglected methane.
Reply: 16% of the study area contain carbonate bedrock. The weathering of the carbonate rock
can be an additional source of DIC in the streams. In our study area, the DIC concentration in the
water increased with the proportion of carbonate containing bedrock in the catchment ($R^2$=0.33,
p<0.001). We added the information on bedrock in the revised version of the manuscript.
On average, DIC in the stream water was composed of 91.2% bicarbonate, 0.4% carbonate and
8.4% $CO_2$. Alkalinity ranged between 0.02 – 13.5 mmol $L^{-1}$ for the individual measurements and
between 0.08 – 9.88 mmol $L^{-1}$ for the averaged seasonal mean values.
Measurements of methane concentration or fluxes are not available for the present study.
According to a recent meta-analysis, the dissolved methane concentration in headwater streams
varies mainly between 0.1 and 1 µmol $L^{-1}$, with streams in temperate forests being at the lower
end (Stanley et al., 2016). The methane makes up only a small fraction of total carbon (in
comparison to the mean DIC concentration in the present study (500 µmol $L^{-1}$)) and we assume
that methane makes a rather small contribution to the catchment scale carbon balance.

We added these information, together with an upper bound of methane evasion (based on the
published meta-analysis), to the revised manuscript.
- Ln 56. Strahler stream order?
Reply: Yes, Strahler stream order. We added this information to the revised manuscript.
- Ln 61-66. What about the geology? Is there C-containing bedrock in the catchments?
Reply: See our comment above - 16% of the study area contain carbonate bedrock. The
weathering of the carbonate rock can be one additional source of DIC in the streams. In our study
area, the DIC concentration in the water increased with the proportion of carbonate containing
bedrock in the catchment ($R^2$=0.33, p<0.001). We added the information on the bedrock in the
revised version of the manuscript.
- Ln 70. "15 800" do not separate numbers
- Ln 71. Delete "order"
Reply: We corrected this.
- Ln 82. How are pH values of investigated waters? The pCO2 calculation with alkalinity
was found to high uncertainties for low pH values (Abril et al. 2014).
Reply: The range of pH values of the investigated waters is 6.2 – 8.97 with a mean of 7.73±0.42
(mean±sd). According to (Abril et al., 2015), high uncertainties of pCO2 estimates from pH and
alkalinity measurements occur at pH values <7, while the median and mean relative errors were
1% and 15%, respectively for pH>7. Only 7% of the pH values in our study were <7. We added
a discussion of the expected uncertainties to the revised manuscript.
- Ln 89. How exactly did you aggregate annual means? Did you calculate a (discharge)
weighted average?
Reply: pCO2_annual=(pCO2_spring+pCO2_summer+pCO2_autumn+pCO2_winter)/4 (seasonal
values are averaged over all available samples). Discharge was not measured during the water
sampling and no time-resolved discharge data are available for the sampling sites. We used
annual mean discharge data, which were derived from data-driven regionalization of discharges
from 125 gauging station from the period of 1979-1998 for the entire fluvial network.
Application of discharge-weighted averaging was therefore not possible.
- Ln 128. Name the program used for statistics
Reply: All statistical analyses were performed with R. We added this information to the revised
manuscript.
- Ln 145-148. Discuss variance of organic C. How about peaty areas?

Reply: There are no pronounced regional or temporal differences of organic carbon. The fraction of peatland in the study area is small (0.95 km$^2$ ; 0.009% of the study area) and only seven of the investigated catchments contain peaty areas. As organic C was measured only in three of these catchments, an investigation of the influence of peat on organic C was not possible.
We added the information about the variance of organic C and the peatland in the study area in the revised version of the manuscript.

- Ln 152. Specify which value is meant: mean NPP or mean specific NPP?
Reply: Mean specific NPP is meant. We clarified this.

- Ln 169. In Figure 3 some of the data points (mostly stream order 1 and one of stream order 2) scatter more. Please discuss reasons for these outliers.

Reply: Small streams of low stream order can be directly influenced by local peculiarities which can increase the scatter of the data points while larger streams represent more average conditions over larger spatial scales. The scattering points in Figure 3 belong, e.g. to ditches or outflows from ponds which might differ in their characteristics to other rivers and streams. Based on the available data, however, there are no particular properties of all the scattering points, which would justify special treatment: the catchments are completely included, pH values are in the range of 7.1-8.3, and no urban areas around these catchments

Ln 186. You talk about average fluxes, but what happens during floods/ extreme events? Do measurement intervals cover extreme events?

Reply: Since we do not have time-resolved discharge data we cannot account for extreme events. Moreover, no information are available if the governmental monitoring included sampling during floods. Given the stochastic nature and short duration, we expect that such samples are at least underrepresented. Since it has been observed that high-discharge events can make a disproportionally high contribution to annual mean carbon export from catchments, we consider our estimates as a lower bound – in accordance with other uncertainty estimates, see below. We added this information to the discussion in the revised manuscript.

- Ln 205. ": : :wetlands covering up to 16 % of the land surface area." Add a reference.

Reply: The 16% was taken from (Richey et al., 2002). However, in the revised version we refer to a fraction of 14%, which was estimated in the study of (Abril et al., 2013). This reference cited earlier in that sentence was moved to the end of the sentence in our revisions.

- Ln 226. Expected for temperate zones? In dry regions such as deserts this can be different.

Reply: We changed the sentence as follows:
*As expected for temperate zones, large streams and rivers with large surface area have larger catchments.*

Reply: For a comparable methodological approach, (Butman and Raymond, 2011), estimated the
uncertainty in the calculation of the aquatic carbon flux to be 33% (based on Monte Carlo
simulation). (Raymond et al., 2013) estimated uncertainties from comparisons of estimates
obtained using similar approaches as we with direct measurements of CO2 concentration on
streams. For a density of sampling locations of 0.02 sites per $km^2$ (corresponding to our study)
they derived an uncertainty of 30%.
In addition to errors associated with sampling and interpolation, our estimates are subject to a
number of systematic errors. The neglect of carbon burial in sediments, carbon export and
evasion as methane and unresolved flood events can be expected to result in an underestimation
of the carbon exported from the catchments in our study. We discussed these uncertainties at
greater detail in the revised manuscript.

Reply: Here we refer to the potentially controlling factors listed in Table 3, including catchment
NPP, fractional water coverage as well as size and climatic zone of the study area. In the revised
manuscript, we listed these factors in the text.

Reply: We agree. 16% of the study area contain carbonate bedrock and the observed association
between DIC and the proportion of carbonate containing bedrock in the catchment ($R^2$=0.33,
p<0.001) indicates, that weathering can be one additional source of DIC in the streams. We
added these results and revised the discussion accordingly. We mentioned that both mineral
weathering and soil respiration contribute to DIC in stream water and discussed the relative
contributions of both sources observed in other studies (Hotchkiss et al., 2015;Lauerwald et al.,
2013;Humborg et al., 2010;Jones et al., 2003).
Examples for cases with predominance of aquatic respiration can not only be found in the
tropics, but also in the boreal zone and in peat-draining streams. We would refer to (Duarte and
Prairie, 2005;Jonsson et al., 2007;Lynch et al., 2010;Richey et al., 2002) as examples.
(Hotchkiss et al., 2015) observed an increased CO2 emissions from internal production for
increasing stream size.

Reply: The range of discharge in our study is 0.003 - 12.2 $m^3$ $s^{-1}$ which is indeed a lower range
compared to the study by Hotchkiss et al. We added the range of discharge in the revised
manuscript.
- Ln 252. Does "small number of observations" relate to this study?
Reply: No, at this point we refer to the meta-analysis presented in Table 3. We made this clearer
in the revised manuscript.
- Ln 255-257. This section summarizes the paper well but it could go further. It might be
speculative but can you say what these other, poorly explored processes could be?
Reply: We would speculate, that hydrology plays a major role for C-cycling at larger scale.
Precipitation controls not only terrestrial NPP but also, drainage density, export of OM from land
to water and retention time of OM in soil and in surface waters respectively. Since our
hydrological data base is rather weak (annual discharge only, no precipitation), we think that
these speculations would be not well supported by the presented results.
- Ln 267. I think it is preferable to provide data as supplement material.
Reply: We provide information about the investigated streams (stream order, water surface area,
discharge, pH), catchment size and catchment NPP, DIC, DOC and TOC, pCO2 and the
seasonality of pCO2, gas exchange velocity and total stream evasion as supplemental material.
- Table 1 and Table 2 would be more informative if you could add ranges. Please also add
calculated gas transfer velocity values to Table 2.
Reply: We added the information to the tables.

where uncertainties were derived based on comparisons of estimates obtained using similar
approaches as we used with direct measurements of $CO_2$ concentration. For a density of
sampling locations of 0.02 sites per km2 they derived an uncertainty of 30%.
Similarly, (Butman and Raymond, 2011) estimated uncertainties of overall flux estimates of
33%, based on Monte Carlo simulation of similar data for hydrographic units in the United
States.
In addition to errors associated with sampling and interpolation, our estimates are subject to a
number of systematic errors. The neglect of carbon burial in sediments, carbon export and
evasion as methane and under-sampling of high-discharge events, probably result in an underestimation of the carbon exported from the catchments in our study. We discussed these
uncertainties at greater detail in the revised manuscript.
- What do you mean "interpolating pCO2 for all river segments without direct measurement"
(Page 4 Line 95-97)? Please clarify in the text.
Reply: The explanation is provided in the following sentence (Line 97-98):
*For this, the mean concentrations were averaged by stream order and assigned to all stream*
*segments of the river network (Butman and Raymond, 2011).*
To clarify this, we joined the two sentences in the revised manuscript.

- For DOC, there are 64 observations (Table 1) in 54 sampling sites (Page 3 Line 90-
91). On average, there are less than 2 observations in each site. Usually, DOC concentrations in
rivers could vary seasonally with river discharge by couples of times.
Therefore, the representativeness of the single DOC data in each catchment remains
a critical question which may induce the great deviation of DOC flux estimation from
the real value. Before resolving this issue, the statements that DOC load only made up
4% total carbon load (Page 5 Line 146-148) and that the error would be comparably
small when neglecting the DOC term (Page 6 Line 159-162) seem arbitrary.
Reply: 54 is a mistake in writing, which we corrected. There are seasonally averaged DOC
observations at 64 sampling sites; all values are averaged over several measurements covering all
seasons.

The authors extensively discuss the aquatic carbon export/NPP ratio in the manuscript
(See Table 3 and text in Section 4.1s). They state in the manuscript:" By combining
CO2 evasion and downstream C-export by stream discharge, we estimated that 2.7
% of terrestrial NPP (13.9 g C m2 yr-1) are exported from the catchments by streams
and rivers, in which both evasion and discharge contributed equally to this flux (Page 7
Line 193-195)". Then they compare their results with some other studies of catchment
ecosystems (see text in Section 4.2). However, what I understand is riverine DIC export
flux is closely related to the weathering regimes and intensity in catchments (See Cai,
W.-J., Guo, X., Chen, C.-T.A., Dai, M., Zhang, L., Zhai, W., Lohrenz, S.E., Yin, K.,
Harrison, P.J., Wang, Y., 2008. A comparative overview of weathering intensity and
HCO3- flux in the world's major rivers with emphasis on the Changjiang, Huanghe,
Zhujiang (Pearl) and Mississippi Rivers. Continental Shelf Research 28, 1538-1549;
and Raymond, P.A., Bauer, J.E., Caraco, N.F., Cole, J.J., Longworth, B., Petsch, S.T.,
2004. Controls on the variability of organic matter and dissolved inorganic carbon ages
in northeast US rivers. Marine Chemistry 92, 353-366) although NPP could contribute
part of DIC export flux through the respiration of DOM. Therefore, the aquatic carbon
export/NPP ratio would be expected to be larger than the real contribution of NPP.

Reply: Our analysis of aquatic C-export in relation to NPP was inspired by studies where
correlations between aquatic C export and terrestrial NPP or NEP have been observed at
different spatial scales and different landscapes (e.g., (Butman et al., 2015;Maberly et al., 2013);
and other studies listed in Table 3). The reason for a lack of correlation in our study could be
related to weathering, as pointed out by the reviewer. 16% of the study area contain carbonate
bedrock. In our study area, the DIC concentration in the water increased with the proportion of
carbonate containing bedrock in the catchment ($R^2$=0.33, p<0.001). In the discussion of the
revised manuscript, we listed this as an uncertainty and added estimates of the contribution of
weathering found in other studies.

the correct formulas would be w=12.88*Q^0.42 and d=0.4*Q^0.29. This corresponds to
width=c*Q^d and depth=a*Q^b. The coefficients and exponents of your equation lead
to depth>width, which seems not realistic.
Reply: Thank you for pointing this out. Indeed the numbers were swapped in the manuscript. All
calculations were done with the correct equations. We corrected the coefficients in the revised
manuscript.

**Anonymous Referee #3**

Katrin Magin and colleagues presented a synthesis of >200 catchments examining the relationships between lateral carbon export and CO2 emissions and terrestrial net primary production (NPP) in southwest Germany. Inland waters have recently been recognized as important components in the global carbon cycle. While widespread studies have been conducted worldwide, most of these studies are based on individual catchments and a synthesis involving multiple catchments remains lacking. This manuscript is well-organized and quite timely, and will provide insights into the under-standing of catchment carbon cycle (or budget) at regional scales.

We would like to thank the reviewer for her/his positive evaluation and the very helpful comments and suggestions. Below we reply on each specific comment.

My first major concern after reading this manuscript is the carbon storage term which has not yet been considered when the authors evaluated catchment-scale carbon budget. Caron burial associated with soil erosion and sediment deposition within catchments is a quite important component in carbon budget assessments (e.g., Smith et al., 2001). If the traditional sediment delivery ratio of 10% is assumed (Harden et al.1999), 90% of the eroded POC from land may have been stored somewhere within the catchment and partly exposed to decomposition (thus evasion to the atmosphere). This missing term may affect the redistribution of carbon (downstream discharge vs. CO2 evasion) as well as the amount of total carbon input from land. Incorporating this term will thus refine the budget result.

Reply: See our response to your specific comment below.

My second concern is the estimation of CO2 evasion. What are the resulting k600 values? Are they comparable to those based on field direct measurements (e.g., floating chamber or eddy covariance)? Estimation of the total areal extent of water surface by means of the parameters derived from USA catchments is probably problematic (see my specific comment below). In addition, can the available dataset suggest any seasonal variability in CO2 evasion?

Reply: The k600 values in our study range from 2.0 m d-1 to 20.6 m d-1 with a mean of 6.0±3.3 m d-1. These transfer velocities are comparable to k600 values based on direct field measurements by floating chambers from small headwater streams in Alaska (Crawford et al., 2013) and also to some short chamber deployments within the study area (Lorke et al., 2015). The pCO2 is higher in summer (mean±sd: 2780±2098 ppm) and autumn (mean±sd: 2848±2019 ppm) than in winter (mean±sd: 2287±1716 ppm) and spring (mean±sd: 2172±2343 ppm). In contrast, the relationship of catchment NPP and CO2 evasion is not influenced by the season. We added the k600 values in Table 2 and discussed the seasonal variability of pCO2 in the revised version of the manuscript.

- Line 19: please clarify 'catchment-specific total export rate'. Is it the normalized carbon export by catchment area?

Reply: Yes, the catchment-specific carbon export rate refers to the carbon export per catchment
area. We clarified this in the revised version of the manuscript.

- Line 29-30: the latest CO2 evasion from global rivers and streams is 0.65 Pg C/yr by
Lauerwald et al., (2015).
Reply: We included the reference in this section.
- Line 50: remove 'differ'.
- Line 71: the reference 'Strahler, 1957' should move to line 56.
- Line 77: remaining →retained
Reply: We applied these corrections.
- Line 81-83. What's the data quality and what kinds of standards for water sampling and
processing were used? Estimating pCO2 from alkalinity and pH has been criticized for
causing biases due to noncarbonate impacts (Abril et al., 2015). An uncertainty analysis
should be provided here. I also suggest to provide the range of pH and alkalinity, possibly into
Table 1.
Reply: The data are governmental monitoring data which are acquired according to DIN EN ISO
norms ((DIN EN ISO 10523:2012-04;DIN EN ISO 9963-1:1996-02;DIN EN ISO 9963-2:1996-
02)). The range of pH values of the investigated waters was 6.2 – 8.97 with a mean of 7.73±0.42
(mean±sd). The range of alkalinity is 0.08 – 9.88 mmol L$^{-1}$ with a mean of 2.75±2.12 mmol L$^{-1}$
(mean±sd). We added pH values and alkalinity in Table 1.
According to (Abril et al., 2015), high uncertainties of pCO2 estimates from pH and alkalinity
measurements occur at pH values <7, while the median and mean relative errors were 1% and
15%, respectively for  pH>7. Only 7 % of the pH values in our study were <7. We added a
discussion of the expected uncertainties to the revised manuscript.

- Line 95-97: how was the site-specific pCO2 interpolated to the upstream catchments?
And which interpolation technique was used?
Reply: The explanation is provided in the following sentence (Line 97-98):
*For this, the mean concentrations were averaged by stream order and assigned to all stream*
*segments of the river network (Butman  and  Raymond,  2011).*
To clarify this, we joined the two sentences in the revised manuscript.

- Line 102-103: These arbitrary parameters derived from American rivers may not necessarily
be representative of German rivers. See Leopold and Maddock (1953).

Reply: The coefficients we used were derived from various data sets obtained in North America,
but have been applied also in global studies before, e.g. (Raymond et al., 2013). Unfortunately,
we are not aware of a comparably extensive data set of hydraulic geometry data derived for
European rivers. A comparison of hydraulic geometry coefficients derived from various data
sets, including data from England, Australia and New Zealand, is presented in (Butman and
Raymond, 2011), who estimated that the error associated with uncertainties of hydraulic
geometry coefficients is rather small, compared to uncertainties derived for C-fluxes. We added
these information to an extended discussion of uncertainties in the revised manuscript.

- Line 105: Is a resolution of 10 m enough to estimate channel slope changes?

Reply:  (Zhang and Montgomery, 1994) investigated the effect of digital elevation model (DEM)
resolution on slope calculation and performance in hydrological models for spatial resolutions
between 2 and 90 m. They found that while a 10-m grid is a significant improvement over 30 m
or coarser grid sizes, finer grid sizes provide relatively little additional resolution. Thus a 10-m
grid size presents a reasonable compromise between increasing spatial resolution and data
handling requirements for modeling surface processes in many landscapes. We justified the
choice of DEM resolution in the revised manuscript.
It should be noted, that similar studies have derived slope information from coarser DEM
resolution, e.g. SRTM 90m Digital Elevation Data in (Lauerwald et al., 2013), GMTED2010
with >250 m resolution in (Raymond et al., 2013), NHDPlus with 30 m ground resolution in
(Butman et al., 2015).

- Line 125-126: Because the mean NPP for the period 2000-2013 is used here while
the pCO2 data is for the period 1970-2011, it is better to explicitly indicate the distribution
frequency of pCO2 data over the study period. For example, if the most of the
pCO2 data were for the period 1970-1980, then using the NPP for 2000-2013 would
be problematic.
Reply: The sampling frequency was increasing. There are nearly twice as much data from 2000-
2011 than from 1977-1999. A comparison between DIC data from both sampling periods
revealed no significant differences. We added the sampling frequency distribution as
supplementary material.
- Line 135: Based on the given definition, the 'drainage rate' term should be 'runoff
depth' in a formal way.
Reply: We corrected the term in our revisions.
- Line 146: Please quantify 'only a small fraction'.

Reply: On average 8.6% of the TOC consist of POC. The highest percentage of POC found in a
catchment is 28.2%.

- Line 158: For the total C input, how about the POC term and the carbon storage term?
See my major comment.

Reply: POC as suspended load in the rivers was estimated along with DOC and was only 8.6%
of the TOC load (0.8 % of the total C-load) at the sampling sites. We agree with the reviewer
that storage can make a significant contribution to the catchment-scale C balance. Estimates vary
between 22% at a global scale (Aufdenkampe et al., 2011), 14 % for the Conterminous U.S.
(Butman et al., 2015) and 39% for the Yellow River network (Ran et al., 2015). However, C
storage in aquatic systems occurs mainly in lakes and reservoirs, which are virtually absent in the
catchments studied here. Therefore we consider the bias caused by neglecting storage to be
comparable in magnitude to remaining uncertainties (30%). We added a more detailed discussion
of the storage term and the associated uncertainty as part of a general uncertainty analysis (see
comments above) in the revised discussion section. The neglect of storage, and potential high C
loads during extreme discharge events, suggest that C-export from catchments estimated in the
present study provides a lower bound of the aquatic C flux.

- Line 220-223: Are there peatlands within the studied catchments?

Reply: The fraction of peatlands in the area in really small (0.009%) and is restricted to 7 of the
investigated catchments. For only 3 of these catchments DOC measurements were available and
no influence of the peatland on the DOC was observable. We added the information about the
peatland in the study area in the revised version of the manuscript.

- Line 229-230: Is the absence of the carbon yield and NPP correlation due to failure
to measure pCO2 during flooding periods? The short-duration carbon export during
flooding events usually accounts for disproportionately a large share of the annual total
carbon export.

Reply: Since we do not have time-resolved discharge data we cannot account for extreme events.
Moreover, no information are available if the governmental monitoring included sampling during
floods. Given the stochastic nature and short duration, we expect that such samples are at least
underrepresented. Since it has been observed that high-discharge events can make a
disproportionally high contribution to annual mean carbon export from catchments, we consider
our estimates as a lower bound – in accordance with other uncertainty estimates, see below. We
added this information to the discussion in the revised manuscript.

- Line 238: please clarify the 'surface area'. The global surface area?

Reply: No, this surface area refers to regions in the 2 studies cited in that sentence. We clarified
this in our revisions.
- Line 244-247: Could it also be because of chemical weathering and groundwater inputs?
Rock weathering in carbonate-dominated catchments can be a significant contributor to DIC. I
would suggest the authors to make a brief introduction about the
lithology and mineralogy in the study area section (2.1).
Reply: 16% of the area investigated in our study contain calcareous bedrock. The DIC
concentration in the water increased with the proportion of carbonate bedrock in the catchments
($R^2=0.33$, $p<0.001$). We added the information about the bedrock in the study area section and
included the influence of chemical weathering in our discussion.
- Line 262: please summarize the study and make a short conclusion.
Reply: We added the following conclusion to the revised version of the manuscript:
*Our analysis of the carbon budget in a temperate stream network on regional scale revealed a*
*relationship of aquatic carbon export and terrestrial NPP. On average 2.7% of the terrestrial*
*NPP were exported from the catchments by rivers and streams with CO2 evasion and*
*downstream transport contributing equally to the export. A comparison of our regional scale*
*study with other studies from different scales and landscapes showed a relatively narrow range*
*of variability of carbon export per catchment area. Future research is needed to understand the*
*processes that control the aquatic-terrestrial coupling and the role of inland waters in regional*
*carbon cycling.*
- Figure 2. It seems the top 2(?) data points far away from the majority are outliers.
Please check and make the regression again, if necessary.
Reply: The apparent outlier is only 1 data point. It does not influence the regression.
- Table 1. pH could also be tabulated here. Is there any trend in pH from SO1 to SO4?
Reply: We included pH in Table 1. There is no trend in pH from SO1 to SO4 but the variability
of pH (e.g. standard deviation) is decreasing with increasing stream order.

[revised manuscript text omitted]

---

## Author Response (AR2)

The authors have addressed most of raised comments in the revised manuscript with comprehensive explanations in the replies. I would still like to raise several minor issues and hope they would be helpful for the further improvement of the manuscript.

We would like to thank the reviewer for her/his positive evaluation and the very helpful comments and suggestions. Below we reply on each specific comment.

1. The authors add a section to demonstrate the uncertainties of flux estimation which could be as high as 30%. Please quantitatively clarify if such high uncertainty would invalidate the model estimation since I don't see the model sensitivity analysis in the manuscript.

Reply: As the uncertainties of the flux estimation are unbiased (i.e. they vary randomly), they do not change the general results of the model. We added this explanation to the uncertainty analysis.

2. In Table 3, the authors compile a large dataset of carbon export of World Rivers for the purpose of comparison with present study. For the Amazon River, Richey et al. (2002) estimated the CO2 evasion of 1.2 Mg C ha-1 yr-1, which was around 87% of total aquatic carbon export. This is different from the value cited in Table 3 (78 g C m-2 yr-1). Probably I am wrong here.

Reply: We changed the value to 138 g C m-2 yr-1.

3. The authors define the aquatic carbon export/NPP ratio as the portion of terrestrial NPP exported by rivers (See P6 Line 192) and frequently make similar statements based on the definition in the sections of Abstract, Discussions and Conclusion. While the NPP and NEP are the main controlling factors regulating the aquatic carbon export, the export/NPP ratio doesn't directly mean that the certain fraction of NPP is exported by rivers. Instead, it only indicates the aquatic carbon export flux represents certain amount of NPP.

Reply: We changed the definition in line 192 to "the ratio of the carbon exported through the aquatic network (i.e. the sum of evasion and discharge) to the terrestrial NPP" and the statements in Abstract, Discussion and Conclusion to "13.9 g C m$^{-2}$ yr$^{-1}$, corresponding to 2.7 % of terrestrial NPP".

4. While this study presents a representative investigation of temperate watersheds, which are a critical component of terrestrial ecosystems and carbon cycling. I would suggest the authors to address the implications of this study in a broad background of global carbon cycling and to clearly unravel their contributions to improve the understanding of the roles of temperate inland water bodies in the regional and global carbon cycles.

[revised manuscript text omitted]